# A single inactivating amino acid change in the SARS-CoV-2 NSP3 Mac1 domain attenuates viral replication *in vivo*

Taha Y. Taha[1,2☉], Rahul K. Suryawanshi[1,2☉], Irene P. Chen[1,2,3☉], Galen J. Correy[2,4☉], Maria McCavitt-Malvido[1,2], Patrick C. O'Leary[2,5], Manasi P. Jogalekar[2,5], Morgan E. Diolaiti[2,5], Gabriella R. Kimmerly[1], Chia-Lin Tsou[1], Ronnie Gascon[1], Mauricio Montano[1], Luis Martinez-Sobrido[2,6], Nevan J. Krogan[2,7,8,9], Alan Ashworth[2,5]*, James S. Fraser[2,4]*, Melanie Ott[1,2,3,10]*

1 Gladstone Institute of Virology, Gladstone Institutes, San Francisco, California, United States of America, 2 Quantitative Biosciences Institute (QBI) COVID-19 Research Group (QCRG), San Francisco, California, United States of America, 3 Department of Medicine, University of California, San Francisco, California, United States of America, 4 Department of Bioengineering and Therapeutic Sciences, University of California, San Francisco, California, United States of America, 5 UCSF Helen Diller Family Comprehensive Cancer Center, San Francisco, California, United States of America, 6 Texas Biomedical Research Institute, San Antonio, Texas, United States of America, 7 Gladstone Institute of Data Science and Biotechnology, Gladstone Institutes, San Francisco, California, United States of America, 8 Quantitative Biosciences Institute (QBI), University of California, San Francisco, California, United States of America, 9 Department of Cellular and Molecular Pharmacology, University of California, San Francisco, California, United States of America, 10 Chan Zuckerberg Biohub–San Francisco, San Francisco, California, United States of America

☉ These authors contributed equally to this work.

* alan.ashworth@ucsf.edu (AA); jfraser@fraserlab.com (JSF); melanie.ott@gladstone.ucsf.edu (MO)

**Data Availability Statement:** All data supporting the findings of the present study are available in the article and supplementary figures and tables.

## Abstract

Despite unprecedented efforts, our therapeutic arsenal against SARS-CoV-2 remains limited. The conserved macrodomain 1 (Mac1) in NSP3 is an enzyme exhibiting ADP-ribosylhydrolase activity and a possible drug target. To determine the role of Mac1 catalytic activity in viral replication, we generated recombinant viruses and replicons encoding a catalytically inactive NSP3 Mac1 domain by mutating a critical asparagine in the active site. While substitution to alanine (N40A) reduced catalytic activity by ~10-fold, mutations to aspartic acid (N40D) reduced activity by ~100-fold relative to wild-type. Importantly, the N40A mutation rendered Mac1 unstable *in vitro* and lowered expression levels in bacterial and mammalian cells. When incorporated into SARS-CoV-2 molecular clones, the N40D mutant only modestly affected viral fitness in immortalized cell lines, but reduced viral replication in human airway organoids by 10-fold. In mice, the N40D mutant replicated at >1000-fold lower levels compared to the wild-type virus while inducing a robust interferon response; all animals infected with the mutant virus survived infection. Our data validate the critical role of SARS-CoV-2 NSP3 Mac1 catalytic activity in viral replication and as a promising therapeutic target to develop antivirals.

**Funding:** This work was supported by the National Institutes of Health (NIAID Antiviral Drug Discovery (AViDD) grant U19AI171110 to N.J.K, F31 grant AI164671-01 to I.P.C.). We gratefully acknowledge support from the Roddenberry Foundation (M.O.), P. and E. Taft (M.O.) and the Pendleton Foundation (M.O.). M.O. is a Chan Zuckerberg Biohub – San Francisco Investigator. The funders did not play a role in the study design, data collection and analysis, decision to publish, or preparation of the manuscript.

**Competing interests:** T.Y.T. and M.O. are inventors on a patent application filed by the Gladstone Institutes that covers the use of pGLUE to generate SARS-CoV-2 infectious clones and replicons. A.A. is a co-founder of Tango Therapeutics, Azkarra Therapeutics, Ovibio Corporation and Kytarro, a member of the board of Cytomx and Cambridge Science Corporation, a member of the scientific advisory board of Genentech, GLAdiator, Circle, Bluestar, Earli, Ambagon, Phoenix Molecular Designs and Trial Library, a consultant for SPARC, ProLynx, GSK and Novartis, receives grant or research support from SPARC and AstraZeneca, and holds patents on the use of PARP inhibitors held jointly with AstraZeneca from which he has benefited financially (and may do so in the future). The Krogan Laboratory has received research support from Vir Biotechnology, F. Hoffmann-La Roche, and Rezo Therapeutics. N.J.K has financially compensated consulting agreements with Maze Therapeutics, Interline Therapeutics, Rezo Therapeutics, and GEn1E Lifesciences, Inc.. N.J.K is on the Board of Directors of Rezo Therapeutics and is a shareholder in Tenaya Therapeutics, Maze Therapeutics, Rezo Therapeutics, and Interline Therapeutics. All other authors declare no competing interests.

## Author summary

The coronavirus disease 2019 (COVID-19) pandemic is caused by severe acute respiratory syndrome coronavirus 2 (SARS-CoV-2) and remains a public health issue worldwide. Despite unprecedented efforts to develop vaccines and therapeutics against SARS-CoV-2, our therapeutic arsenal remains limited to only a limited number of therapeutics due to the lack of validated viral drug targets. Here, we identify a potentially new drug target in SARS-CoV-2. The Mac1 domain is an evolutionarily conserved enzyme in coronaviruses that can remove ADP-ribose post-translational modifications from proteins thereby combating host innate immune responses. We generated SARS-CoV-2 with a single point mutation that renders the Mac1 domain catalytically inactive and demonstrated that this mutant virus is less fit in mini lungs in the dish (organoids) and in mice. This work paves the way for the development of novel therapeutics targeting the Mac1 catalytic activity as SARS-CoV-2 antivirals.

## Introduction

The coronavirus disease 2019 (COVID-19) pandemic continues to be a major public health crisis due to the emergence of severe acute respiratory syndrome coronavirus 2 (SARS-CoV-2) variants with enhanced transmissibility and immune escape [1]. As of April 2023, more than 760 million people have been infected and more than 6.9 million have died of COVID-19 [2]. Despite unprecedented efforts to develop safe and effective vaccines and therapeutics, our arsenal against SARS-CoV-2 remains limited to a few drugs, including remdesivir and molnupiravir targeting the viral RNA-dependent RNA polymerase (RdRp) and nirmatrelvir (ritonavir-boosted as Paxlovid) targeting the viral main protease (MPro). While each of these drugs have shown efficacy in clinical studies [3–5], each is associated with certain drawbacks such as administration via intravenous route for remdesivir, rise of mutations with molnupiravir [6], and drug-drug interactions and resistance mutations of Paxlovid [5,7–9]. There is a critical need to develop novel antivirals targeting other viral proteins to facilitate combination, potentially synergistic, therapeutics and minimize development of drug resistance. Therefore, identification and validation of viral proteins other than the RdRp or MPro as therapeutic targets is necessary.

SARS-CoV-2 is a positive-stranded RNA virus in the *coronaviridae* family and the *betacoronavirus* genus, which also includes SARS-CoV, Middle East respiratory syndrome coronavirus (MERS-CoV), and murine hepatitis virus (MHV). The virus encodes several open reading frames including two large polyproteins 1a and 1ab (PP1a and PP1ab). These polyproteins contain 16 non-structural proteins (NSPs) that are essential components of the viral replicase complex. NSP3 is the largest viral protein and contains several domains including three macrodomains: Mac1, Mac2, and Mac3 [10]. Macrodomains are ubiquitous in nature across eukaryotes, prokaryotes, and archaea [11]. While Mac2 and Mac3 are catalytically inactive, Mac1 can bind ADP-ribosylated protein sidechains and catalyze the hydrolysis of these post-translational marks for a diverse set of protein targets [10,12–14]. Macrodomains are found and conserved in over 150 viruses [11] and are structurally [15] and evolutionarily [16] related to cellular macrodomains, albeit with distinct functions [17]. Viral macrodomains play critical roles in the replication of alphaviruses and hepatitis E virus [18,19]. In coronaviruses (CoV), Mac1 within NSP3 is structurally conserved in several betacoronavirus lineages including lineage A (MHV), lineage B (SARS-CoV and SARS-CoV-2), and lineage C (MERS-CoV) [20] (S1

Fig). Collectively, the acquisition, conservation, and evolution of viral macrodomains suggests an evolutionarily conserved fitness advantage of these proteins.

The Mac1 of coronaviruses, including SARS-CoV-2, has ADP-ribosylhydrolase activity *in vitro* [20]. ADP-ribosylation is a ubiquitous post-translational modification that is "written" by poly-adenosine diphosphate-ribose polymerases (PARPs) and is associated with a diverse array of biological consequences (reviewed in [21,22]), including DNA repair, signal transduction pathways, epigenetics and transcription, and many others. Importantly, PARPs have been shown to be important regulators of virus-host interactions and are integrated in many interferon (IFN) signaling pathways (reviewed in [23]). PARP7, PARP9, PARP12, and PARP14 directly enhance antiviral innate immune responses [24–26] and are upregulated during MHV infection [27]. PARP7 has also been shown to play a proviral role in influenza A virus infection [28]. Notably, PARP14-induced ADP-ribosylation is reversed by SARS-CoV-2 Mac1 [29]. The acquisition of an ADP-ribosylhydrolase such as Mac1 in SARS-CoV-2 NSP3 can therefore be seen as a means for the virus to "erase" ADP-ribosylation marks and dampen the antiviral innate immune response. Indeed, previous studies in coronaviruses have demonstrated that the NSP3 Mac1 is necessary for pathogenesis and for robust viral replication [30–33].

Previous studies have typically employed a mutant virus where the conserved asparagine residue at position 40 in the Mac1 active site (S1 Fig) is mutated to an alanine to render NSP3 Mac1 inactive. In MHV, the Mac1-deficient virus replicated to similar levels as the wild-type (WT) virus *in vitro* but was significantly attenuated in primary bone marrow-derived macrophages and in mice [30,31]. Interestingly, infection with Mac1-deficient MHV-A59 (hepatotropic) was associated with reduced IFN induction whereas infection with Mac1-deficient MHV-JHM (neurotropic) induced higher levels of IFN in mice [30,31]. The difference between two MHV strains that cause distinct disease pathologies indicates an important role in the host cell type in determining the function of the MHV Mac1. Similar experiments were conducted in SARS-CoV and showed comparable results where the mutant virus displayed replication defects only in mice, but not in cell culture models, and was associated with higher induction of IFN [32]. Collectively, these studies suggest a critical role for coronavirus NSP3 Mac1 in viral replication that may not be fully recapitulated in immortalized cell lines. SARS-CoV-2 NSP3 Mac1 has ~75% sequence conservation with SARS-CoV (S1 Fig) and has been shown to catalyze the hydrolysis of ADP-ribose moieties from a diverse set of targets [13,20]. Recently, Alhammad et al. generated a SARS-CoV-2 Mac1 deletion mutant (ΔOrf1a 1023–1192) and showed that the mutant has significantly attenuated IFN antagonism and replication *in vivo* [34]. The role of the NSP3 Mac1 catalytic activity in SARS-CoV-2 replication and pathogenesis remains unknown.

Here, we show that SARS-CoV-2 NSP3 Mac1 catalytic activity is necessary for robust viral replication *in vivo* and is a reasonable target for the development of novel therapeutics. To catalytically inactivate SARS-CoV-2 Mac1, we focused on mutating the conserved asparagine residue at position 40. Biochemical analysis of alanine and aspartic acid mutants at this residue demonstrated that while both mutants lack catalytic activity, the N40A mutant Mac1 is destabilized and inherently unstable in bacterial and mammalian cells. Using reverse genetics, we show that a SARS-CoV-2 replicon containing NSP3 Mac1 N40A mutation is slightly attenuated in cell culture models without affecting IFN induction, while a replicon and infectious virus containing N40D mutation were associated with higher IFN induction but show minimal replicative effects in immortalized cell lines. In primary airway organoids and in mice, the N40D virus is profoundly attenuated demonstrating the same dichotomy of phenotypes between *in vitro* and *in vivo* studies of Mac1-deficient CoVs. Our studies demonstrate that SARS-CoV-2 Mac1 is a critical viral enzyme for viral replication *in vivo*.

## Results

### Residue N40 is critical for the catalytic activity and protein stability of SARS-CoV-2 Mac1

SARS-CoV-2 NSP3 Mac1 is a MacroD-type macrodomain with high structural conservation with other CoV macrodomains despite sequence divergence (S1 Fig). To determine the role of Mac1 in SARS-CoV-2 replication, we wanted to generate Mac1 catalytic activity-deficient SARS-CoV-2 through point mutations that abrogate Mac1 catalytic activity. Several conserved residues in the active site [20], especially asparagine 40 (N40), have been utilized in previous studies of CoV to generate such mutant viruses [30–33]. Mac1 N40 coordinates the distal ribose moiety in an ADP-ribosylated substrate and is critical for binding and catalytic activity [32,35]. To investigate the structural and functional contribution of N40 to binding and catalytic activity, we generated Mac1 N40A and N40D mutants that are both predicted to disrupt coordination of the distal ribose moiety and therefore significantly inhibit catalytic activity of Mac1. When expressed as an isolated Mac1 domain in bacteria, the soluble yield of N40A mutant was typically 20-fold lower compared with the WT and N40D Mac1 proteins (Fig 1A). This suggests that the N40A mutant is unstable or fails to fold efficiently. To determine the thermostability of the mutants in the absence or presence of ADP-ribose, we conducted differential scanning fluorimetry (DSF) and found that the N40A mutant unfolds at a significantly lower temperature (34˚C) compared with the WT (49˚C) and N40D mutant (46˚C) (Fig 1B and 1C). All variants were significantly stabilized by ADP-ribose, indicating that binding activity is not greatly compromised (Fig 1B and 1C). These data indicate that the N40A mutant may have an inherently unstable fold compared to the WT and N40D mutant, which could be responsible for the low stability and expression yield in bacteria. To determine whether this is also the case in mammalian cells, we generated lentiviral mammalian expression vectors of Strep-tagged isolated Mac1 domain and transduced A549 lung carcinoma cells. Western blot analysis of transduced cells revealed that the N40A mutant is expressed at significantly lower levels compared with the WT and N40D mutant (S2 Fig). Collectively, these data indicate that Mac1 tolerates an N40A mutation poorly, but it can accommodate an N40D mutation. Interestingly, such a profound effect of the N40A mutant on protein stability in the context of the entire NSP3 protein was not observed for SARS-CoV NSP3 Mac1 as it only resulted in ~16% reduction in NSP3 expression [32], which suggests different physicochemical properties of these proteins despite high structural similarity.

To determine the binding pose of ADP-ribose for the Mac1 N40D mutant, we crystalized the N40D Mac1 mutant protein in the presence of ADP-ribose (Fig 1D and 1E). Relative to WT, the ADP-ribose bound conformation is conserved in the mutant, but the terminal dihedral of the D40 side chain is rotated ~80˚ relative to N40 and therefore does not form a hydrogen bond with the terminal ribose. This rotation is also observed in the apo structure of N40D (S3 Fig). Interestingly, in the structure of N40D Mac1 protein bound to ADP-ribose, we observe a water molecule (purple sphere labeled W in Fig 1E) forming a new hydrogen bond to the terminal ribose that mimics the interaction observed to the nitrogen of N40. To determine whether these subtle differences in binding affect Mac1 enzymatic activity, we conducted deMARylation assays with auto ADP-ribosylated PARP10 and the WT and mutant Mac1 proteins (Fig 1F). We found that all proteins were capable of deMARylation but the N40D mutant had lower deMARylation activity (Fig 1F). To quantify the catalytic activity of these proteins, we utilized the NudT5/AMP-Glo assay to measure ADP-ribose released by the enzymatic reaction using adjusted substrate concentrations (Fig 1G). Given the low activity of the mutants, a 10- and 100-fold excess of the N40A and N40D mutants, respectively, was used in the assay relative to the WT Mac1. Relative to WT Mac1 catalytic activity ($k_{cat}/K_M = 0.37 \ \mu M^{-1} \ min^{-1}$), the N40A mutant had <10% activity ($k_{cat}/K_M = 0.029 \ \mu M^{-1} \ min^{-1}$), while the N40D mutant had <1% activity ($k_{cat}/K_M = 0.0026 \ \mu M^{-1} \ min^{-1}$)

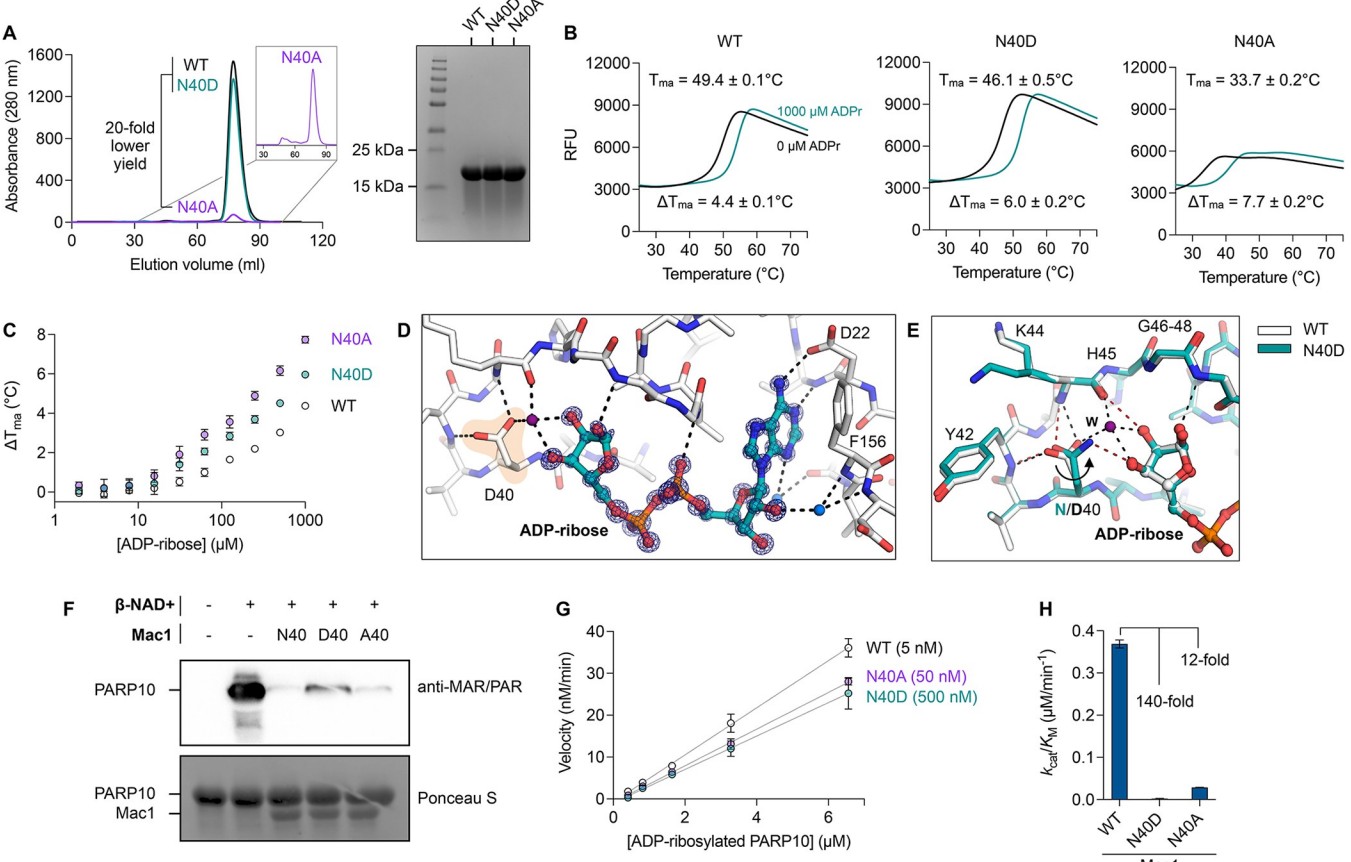

**Fig 1. Expression and characterization of Mac1 N40A and N40D mutants *in vitro*.** A) Analysis of WT, Asn40Ala (N40A) and Asn40Asp (N40D) soluble expression in BL21(DE3) *E. coli* by size exclusion chromatography (HiLoad 16/600 Superdex 75 pg column) and SDS-PAGE (Coomassie stained). B) Thermostability of Mac1 mutants determined by differential scanning fluorimetry (DSF). Protein (2 µM) was incubated ± 1 mM ADP-ribose with SYPRO orange (10 µM). Apparent melting temperatures were calculated by DSFworld [66] using fitting model 1. Data are presented as the mean of four replicates. C) Difference in $T_{ma}$ for Mac1 incubated with ADP-ribose from 2 µM to 1 mM. Data are presented as the mean ± SD of four technical replicates. D) X-ray crystal structure of ADP-ribose bound to N40D. Difference electron density ($mF_O$-$DF_C$ map) is shown prior to modeling ADP-ribose (blue mesh contoured at 8 σ). E) ADP-ribose binding is conserved between WT (teal sticks) and N40D (white sticks). Selected hydrogen bonds are shown for WT (red dashed lines) and N40D (black dashed lines). The D40 side chain is rotated ~80° relative to N40 and therefore does not form a hydrogen bond with the terminal ribose, however, a water molecule (purple sphere labeled W) creates a new hydrogen bond network. F) Western blot analysis of Mac1 activity with auto-MARylated PARP10 (residues 819–1007). MARylated PARP10 (0.5 µM) was incubated with Mac1 variants (20 µM) for 1 hour at room temperature prior to SDS-PAGE and transfer to nitrocellulose membrane and blotting with anti-MAR/PAR antibody (1:1000 dilution, Cell Signaling, 83732). G) Kinetic measurements of MARylated PARP10 hydrolysis by Mac1 variants (concentration indicated between parentheses) using NudT5/AMP-Glo to detect ADP-ribose [51,59]. H) Catalytic efficiency ($k_{cat}/K_M$) was determined by linear regression of data in G using GraphPad Prism. Data are presented as the mean ± SD of three technical replicates.

(Fig 1H). These data demonstrate that both mutants have significantly reduced Mac1 enzymatic activities with the N40D mutant having 10-fold and 100-fold lower enzymatic activity relative to the N40A mutant and the WT Mac1, respectively. Taken together, these data suggest that the Mac1 N40D mutant is a good surrogate for abolishing catalytic activity in a similar manner to an inhibitor, whereas the significant destabilization observed with the N40A mutant may render it difficult to interpret in the context of the multi-domain structure of NSP3.

## The Mac1 N40D replicon has reduced ability to suppress innate immune responses in cell culture

Next, we introduced the N40A and N40D mutations into SARS-CoV-2 replicons to determine their impact on viral RNA replication in a rapid and safe manner [36]. The replicons express

secreted nanoluciferase as a reporter *au lieu* of the Spike-coding sequence and can produce single-round infectious particles when supplied with a separate Spike expression vector (Fig 2A). Single-round infectious particles were generated for the ancestral (WA1) replicon and the two Mac1 N40A and N40D mutants. They were used to infect simian Vero cells overexpressing the ACE2 and TMPRSS2 entry factors (VAT) or human Calu3 lung adenocarcinoma cells naturally expressing both factors. We measured secreted nanoluciferase in the supernatant, which reflects intracellular viral RNA levels [36]. The N40D mutant resulted in a small but statistically significant 1.4- and a non-statistically significant 1.1-fold reduction in luciferase signal relative to the WT replicon in VAT and Calu3 cells, respectively, while the Mac1 N40A mutant showed a significant (5- and 13-fold) reduction (Fig 2B). Intracellular RNA was isolated from infected Calu3 cells, and both viral and host RNAs were measured by RT-qPCR. Viral RNA levels were similar between the WT and N40D mutant replicons, while the N40A mutant had a ~44-fold reduction in viral RNA levels relative to the WT replicon (Fig 2C). Measurement of a panel of four innate immune response genes (IFNb, ISG15, IL6, and STAT1) demonstrated higher induction in N40D-infected cells, especially levels of ISG15, IFNb and IL6, as compared to WT and N40A (Fig 2D). No such difference was observed with the N40A replicon despite the slight change in viral replication (Fig 2D). This result suggests that the modest replication deficit but lack of IFN and innate immune response genes induction of the N40A replicon could result from the observed Mac1 instability (S2 Fig) rather than a decrease in catalytic activity (Fig 1H). Therefore, we prioritized the N40D mutant in subsequent experiments to avoid the confounding destabilizing effect of the N40A mutation on Mac1.

To determine if the observed increase in the innate immune response was too weak to suppress the N40D mutant replication, Calu3 cells were treated with varying doses of exogenous IFN gamma or beta (IFNγ or IFNβ, respectively) and then infected with single rounds of WT and N40D replicons. This treatment significantly decreased viral RNA levels of both replicons compared to replication levels in untreated cells as measured by RT-qPCR (Fig 2E and S4 Fig). The 10 and 100 ng/μL doses of IFNγ suppressed the Mac1 N40D replicon by 2.5- and 3-fold, respectively, more than the WT replicon (Fig 2E). At 1000 ng/μL IFNγ and all doses of IFNβ, activity of both replicons was suppressed to similar levels (Fig 2E and S4 Fig). Collectively, these data suggest a modest, but specific, effect of the Mac1 N40D mutation on reversing the suppression of innate immune responses in immortalized cell lines.

## The Mac1 N40D infectious virus has a 10-fold replication deficit in human airway organoids

To determine whether the Mac1 catalytic activity plays a role in a full replicative infection cycle, we introduced the Mac1 N40D mutation into a WA1-based infectious clone. The clone was transfected into BHK-21 cells and then rescued and passaged over Vero TMPRSS2 cells to achieve high viral titers. After sequence verification, the plaque morphology of the viruses was determined using Vero TMPRSS2 cells; the WT and the N40D mutant showed similar plaque morphology (Fig 3A). The viruses were then used to infect Calu3 cells; at 48 hours after infection, particle production was measured by plaque assay while intracellular viral RNA and cytokine expression levels were measured by RT-qPCR. The N40D Mac1 mutant showed modestly (~2-fold) lower particle production and similar intracellular viral RNA levels compared with the WT virus suggesting a slightly attenuated phenotype (Fig 3B and 3C). Particle production was not significantly different between the N40D mutant and WT viruses over 72 hours post-infection and at multiple MOIs, except for mild enhancement of the mutant virus at 72 hours post-infection at MOI 0.1 (S5 Fig). Similar to the results from replicon experiments, the N40D

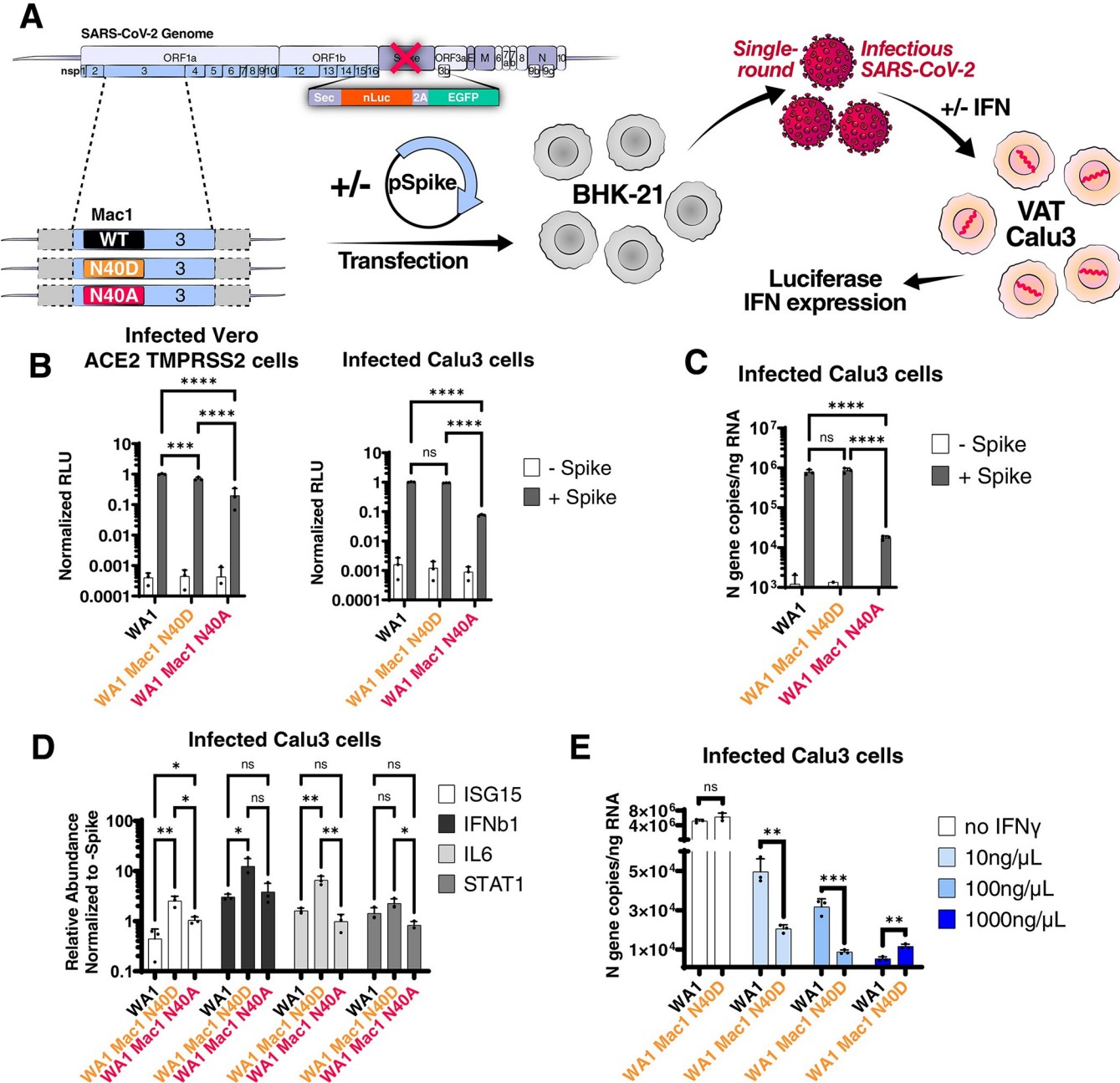

**Fig 2. SARS-CoV-2 replicon with Mac1 N40D mutation fails to suppress innate immune responses in cells.** A) Experimental workflow of the SARS-CoV-2 replicon system. SARS-CoV-2 WA1 and Mac1 domain mutant replicons were transfected along with a S and N expression vectors into BHK-21 cells. At 72 hours post-transfection, the supernatant containing single-round infectious particles is used to infect VAT and Calu3 cells in the presence or absence of exogenous IFN. At 72 hours post-infection, luciferase activity in the supernatants was used as a readout for viral RNA replication and innate immune response gene expression is measured by RT-qPCR. B) Luciferase readout of infected VAT and Calu3 cells with indicated replicons. Data are presented as mean +/- SD of three biological replicates conducted in triplicate. C) Intracellular viral RNA (N gene copies) of Calu3 cells in B was measured by RT-qPCR using a standard curve. Data are presented as mean +/- SD of three biological replicates conducted in triplicate. D) Relative expression of indicated innate immune response genes relative to GAPDH control and normalized to the -Spike control of each replicon in infected cells in B by RT-qPCR. Data are presented as mean +/- SD of three biological replicates conducted in triplicate. E) Calu3 cells were infected with indicated replicons in the presence of indicated IFNγ concentrations. Intracellular viral RNA (N gene copies) was measured by RT-qPCR using a standard curve. Data are presented as mean +/- SD of one biological replicate conducted in triplicate. *, $p < 0.05$; **, $p < 0.01$; ***, $p < 0.001$; ****, $p < 0.0001$ by two-sided Student's T-test.

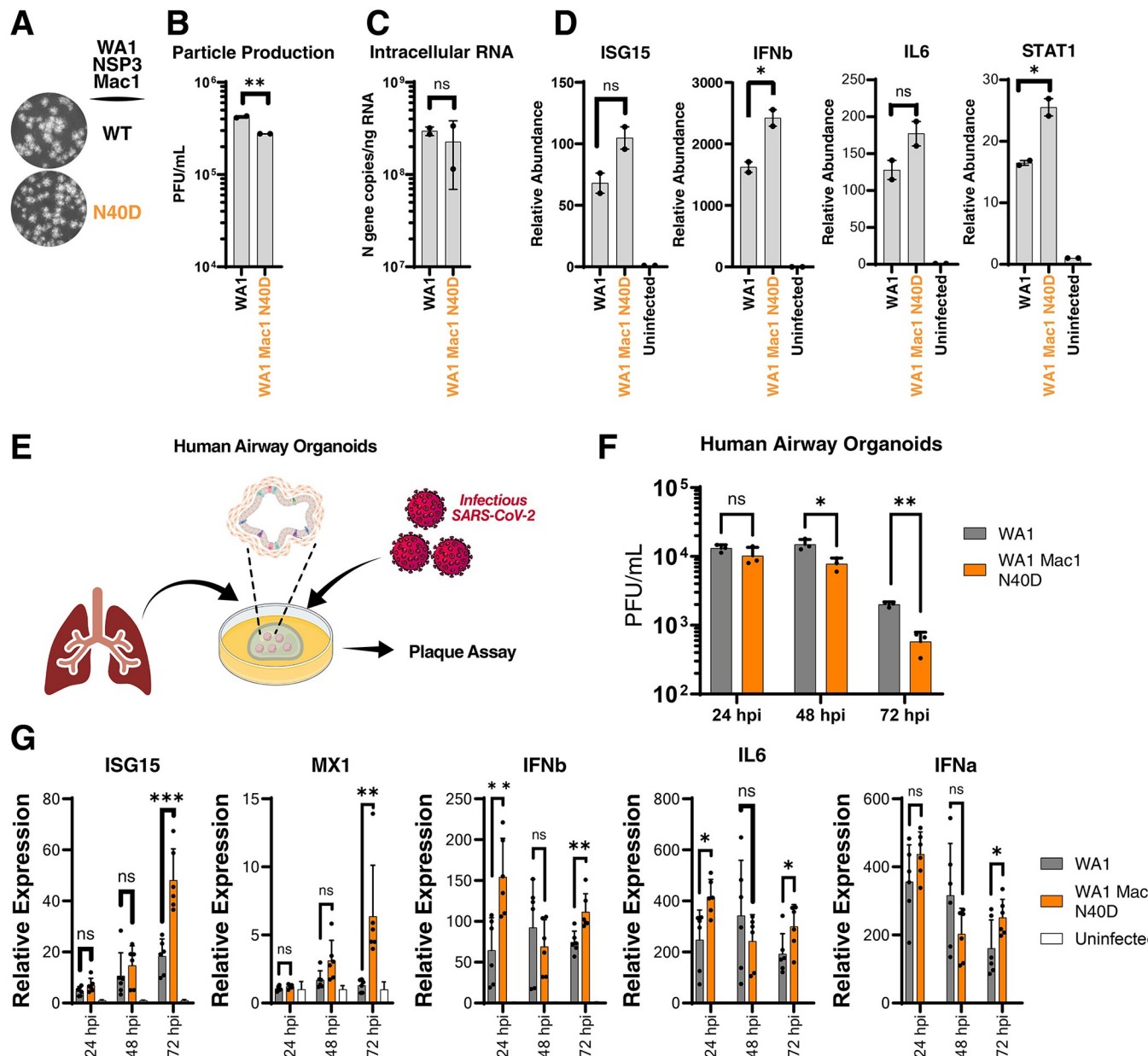

**Fig 3. SARS-CoV-2 Mac1 domain N40D mutant is attenuated in human airway organoids.** A) Plaque morphology of indicated viruses in VAT cells. The images were pseudocolored in grayscale to optimize plaque visualization. B) Viral particle release of infected Calu3 cells (MOI 0.01) with indicated viruses was measured at 48 hours post-infection by plaque assay on VAT cells. Data are presented as mean +/- SD of two biological replicates conducted in duplicate. C) Intracellular viral RNA (N gene copies) of Calu3 cells in B was measured by RT-qPCR using a standard curve. Data are presented as mean +/- SD of two biological replicates conducted in duplicate. D) Relative levels of ISG15, IFNb, IL6, and STAT1 in Calu3 cells in B normalized to uninfected cells were determined by RT-qPCR. Data are presented as mean +/- SD of two biological replicates conducted in duplicate. E) Experimental workflow of the generation of primary human lung organoids and infection with SARS-CoV-2. Clip art was created with [BioRender.com](BioRender.com) with permission. F) Viral particle release of infected primary human lung organoids with indicated viruses was measured at indicated times post-infection by plaque assay on VAT cells. Data are presented as mean +/- SD of one biological replicate conducted in triplicate. G) Relative levels of ISG15, MX1, IFNb, IL6, and IFNa to 18s RNA in infected human airway organoids in F normalized to uninfected organoids were determined by RT-qPCR. Data are presented as mean +/- SD of one biological replicate conducted in triplicate. *, $p < 0.05$; **, $p < 0.01$; ***, $p < 0.001$ by two-sided Student's T-test.

mutant induced higher levels of innate immune response genes as compared to the WT virus (Fig 3D), underscoring the role of Mac1 catalytic activity in innate immune response antagonism.

It is possible that the modest phenotype of the N40D mutant observed in Calu3 cells is due to these cells not fully recapitulating the host environment during SARS-CoV-2 infection. Human lung organoids are stem-cell derived multi-cell type 3D structures that have intact innate immune responses and better recapitulate the environment during SARS-CoV-2 infection [37,38]. We tested the replicative capacity of the N40D mutant in differentiated human airway organoids (Fig 3E). While WT and Mac1 N40D mutant viruses produced similar levels of infectious particles at 24 hours post-infection, the Mac1 N40D mutant showed ~10-fold lower particle production at 48 and 72 hours post-infection (Fig 3F). Moreover, the expression of several innate immune response genes (ISG15, IFNb, IL6, MX1, and IFNa) was higher in organoids infected with the N40D mutant compared with the wild-type virus (Fig 3G). The observed fast viral clearance of the N40D mutant and the concomitant induction of innate immune response genes support the model that loss of IFN antagonism limits viral infection by the N40D mutant virus, especially at later time points after infection. These data are in contrast to previous work with SARS-CoV where the Mac1 catalytic activity-deficient virus replicated at low levels early post-infection compared to the WT virus suggesting potentially attenuated infectivity rather than viral clearance [33]. Collectively, these data demonstrate that SARS-CoV-2 Mac1 catalytic activity plays a critical role in viral replication in primary cells that is not fully recapitulated in immortalized cell lines.

## SARS-CoV-2 Mac1 N40D mutant is significantly attenuated *in vivo*

To ascertain whether SARS-CoV-2 Mac1 catalytic activity plays a role in viral replication *in vivo*, we intranasally infected mice overexpressing the human ACE2 receptor (K18-hACE2) with WT and Mac1 N40D viruses (Fig 4A). In response to WT viral infection, K18-hACE2 mice typically develop significant weight loss and lowered body temperature and reach the humane endpoint by day 6 post-infection due to viral encephalitis [39,40]. In contrast, mice infected with the Mac1 N40D mutant did not experience any changes in weight or body temperature, and all mice survived 6 days post-infection (Fig 4B–4D). This data indicate that the SARS-CoV-2 Mac1 catalytic activity plays an essential role in viral pathogenesis *in vivo*.

When viral particle production and tissue viral RNA levels were examined in the lungs, upper respiratory tract, and brain, the Mac1 N40D mutant had slightly higher particle production and viral RNA levels at 2 days post-infection but significantly less 4 days post-infection and undetectable levels 6 days post-infection in the lungs (Fig 4E and 4F). Similar decreases were observed in the upper respiratory tract albeit with overall lower viral particle production and tissue RNA levels as expected (S6A and S6B Fig). Notably, no infectious particles were recovered from the brain of animals infected with Mac1 N40D mutant virus, consistent with the 100% survival rate of the infected mice (S6C Fig). These data indicate efficient entry and replication early post-infection, but significantly faster clearance of the Mac1 N40D virus compared with the WT virus. These results are also consistent with other coronavirus studies, demonstrating a critical role of the Mac1 domain in viral replication *in vivo* [30–33]. Lung histology of mice infected with the WT and Mac1 N40D virus showed SARS-CoV-2 infection-related pulmonary pathology [41] at 2 days post-infection characterized by inflammation and thickening of alveolar septa by infiltrating immune cells (Fig 4G). Interestingly, the lung pathology for the mice infected with the Mac1 N40D virus appeared mostly similar to that of the mice infected with the wild-type virus (Fig 4G). Independent pathology analysis revealed that the N40D mutant virus had higher pathology score at 2 days post-infection but similar

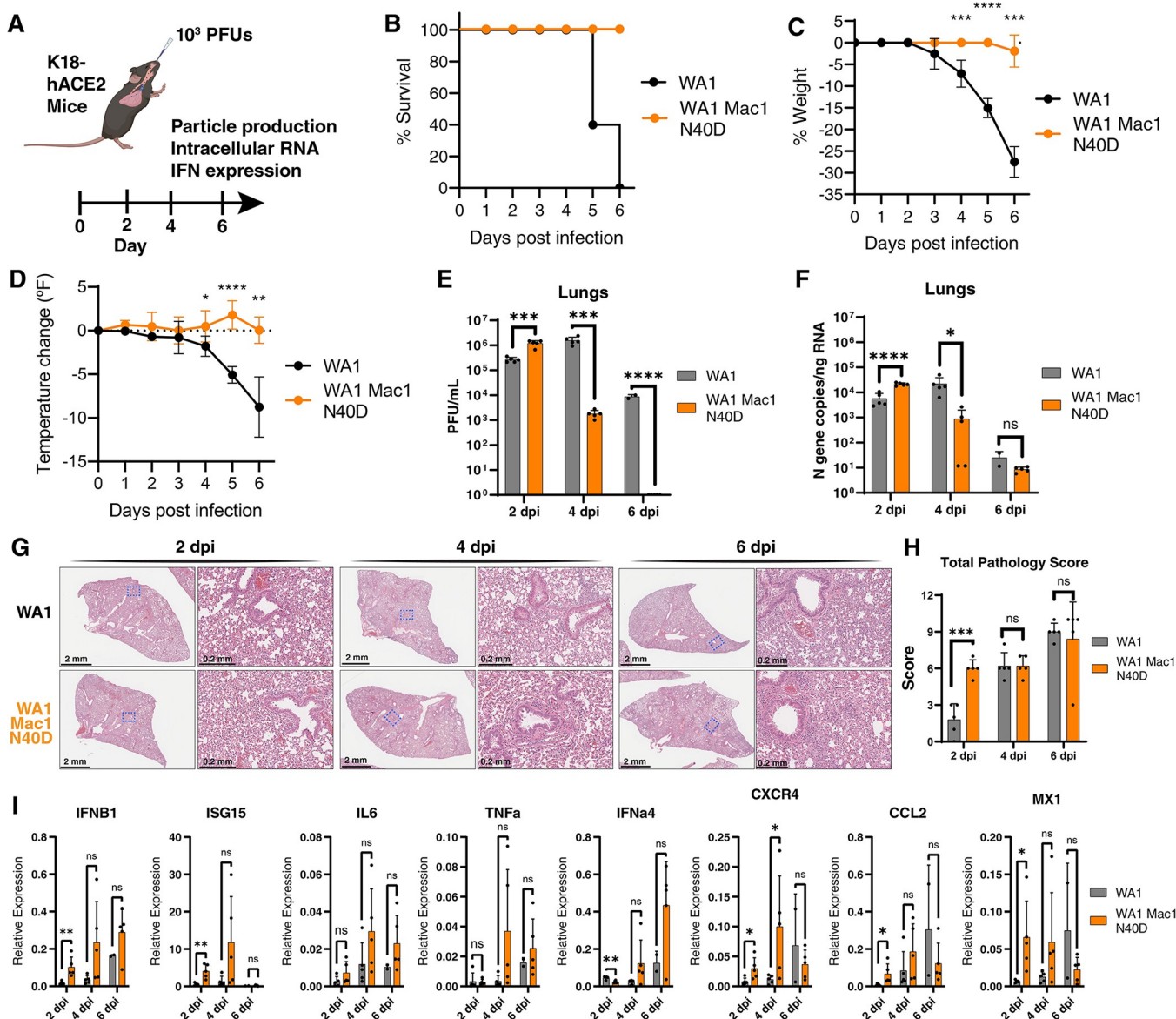

**Fig 4. SARS-CoV-2 Mac1 N40D mutant is attenuated in K18-hACE2 mice.** A) K18-hACE2 mice (5 mice per group per timepoint) were infected with $10^3$ PFUs of WA1 and WA1 Mac1 N40D viruses intranasally. At 2, 4, and 6 days post-infection, the lung tissue was collected for viral particle abundance measurement by plaque assay (VAT) and viral RNA and innate immune response gene expression analysis by RT-qPCR. Clip art was created with BioRender. com with permission. B) Survival curve of mice infected in A. C) Change in body weight of mice infected in A. Data are shown as mean ± SD. D) Change in body temperature of mice infected in A. Data are shown as mean ± SD. E) Viral particle abundance in the lungs was measured by plaque assay (VAT). Data are presented as mean +/- SD. F) Viral RNA levels in the lungs were measured by RT-qPCR using a standard curve. Data are presented as mean +/- SD. G) Hematoxylin and eosin (H&E) staining of lungs of mice infected in A. H) Total pathology scores of all infected mice lungs in G. Breakdown of scores is available in S7 Fig. I) Relative levels of IFNB1, ISG15, IL6, TNFa, IFNa4, CXCR4, CCL2, and MX1 to GAPDH were determined by RT-qPCR. Data are presented as mean +/- SD. *, $p < 0.05$; **, $p < 0.01$; ***, $p < 0.001$; ****, $p < 0.0001$ by two-sided Student's T-test.

scores at 4 and 6 days post-infection (Fig 4H). Specific pathology scoring of the vascular, bronchial and alveolar areas of the lung followed this trend (S7A–S7C Fig), except for alveolar inflammation which was higher in the wild-type virus compared with the N40D mutant (S7D Fig). The extent of inflammation in the different areas of the lung were also quantified and were higher at 4 and 6 days post-infection for the N40D mutant compared with the wild-type virus (S7E–S7G Fig). Collectively, the observed significant differences in survival (Fig 4B) and

viral replication (Fig 4E and 4F) but similar histological findings (Fig 4G, 4H and S7 Fig) support the model that the N40D mutant virus can infect and replicate to robust levels and induce similar lung immune pathology, but is cleared quicker than the wild-type virus.

We also collected lung tissues from WT- and N40D-infected animals and extracted RNA for innate immune response genes expression analysis by RT-qPCR. We selected several genes implicated in SARS-CoV-2 clearance and pathogenesis, including IFNB1, ISG15, IL6, TNFa, IFNa4, CXCR4, CCL2, and MX1 [42]. At all assayed timepoints, Mac1 N40D viral infection was consistently associated with higher innate immune response gene expression relative to the WT virus (Fig 4I). The most highly induced gene was ISG15 at 2 and 4 days post-infection, while IFNa4 was highly induced at day 6 post-infection (Fig 4I). Notably, similar innate immune response gene upregulation was previously observed in SARS-CoV Mac1-deficient virus-infected mice [32]. Collectively, these data support the model that the SARS-CoV-2 Mac1 catalytic activity is an important suppressor of innate immune responses and thus critical for viral replication *in vivo*.

## Discussion

The SARS-CoV-2 Mac1 domain represents an attractive drug target due to its catalytic ADP-ribosylhydrolase activity, its conserved structure, and its relative lack of observed mutations in emerging SARS-CoV-2 variants [43]. Here we identify–through biochemical and structural characterization–a single amino acid change within the active site (N40D) that diminishes the catalytic activity to <1% of WT activity while maintaining protein stability. This mutation, when introduced via reverse genetics into fully infectious viral clones or replicons, showed minimal effects on viral replication in immortalized cell lines while modestly inducing the IFN response. However, in primary lung organoids and in mice, we observed profound effects of the N40D mutation on viral replication and innate immunity induction, demonstrating that Mac1 is a critical virulence factor.

CoV Mac1 contains a conserved structure related to MacroD family of macrodomains [11,35]. Several residues in the active site coordinate the ADP-ribose group and have been shown to be critical for ADP-ribosyl substrate binding and catalysis [27]. One of the most critical residues is N40 which makes hydrogen bonds with the terminal ribose hydroxyl group and is conserved in all CoV macrodomains. Here we show that this position within the SARS-CoV-2 Mac1 domain does not tolerate mutation to alanine due to protein instability. These results contrast with SARS-CoV Mac1 where the A mutant appears to only modestly reduce the expression of NSP3 in infected cells [32]. While structurally conserved, the SARS-CoV-2 Mac1 sequence diverges by 25% from SARS-CoV, and this sequence difference may endow different physicochemical properties to the SARS-CoV-2 Mac1 domain as compared to other CoVs despite maintaining the ADP ribosylhydrolase activity and a well-conserved structural fold.

The CoV Mac1 catalytic activity has been shown to contribute to the suppression of innate immune responses during SARS-CoV and MHV-JHM infection [31,32]. Recently, a SARS-CoV-2 mutant lacking Mac1 expression showed significant attenuation *in vivo* and reduced IFN antagonism [34]. It is not entirely surprising that immortalized cell lines do not fully recapitulate the phenotype of SARS-CoV-2 WT and Mac1 N40D mutant in primary cells or mice. Previous studies with CoV either found no differences in cell culture between the WT and Mac1 catalytic activity-deficient viruses or required the use of *in vivo* models or the screening of several cell culture systems and primary cells to find a cell type in which differences were apparent [30–32,34]. We used two different cell culture lines frequently used in SARS-CoV-2 research: Vero cells (engineered to overexpress ACE2 and TMPRSS2) and Calu3 cells. Vero

cells lack IFN production [44] while Calu3 cells have intact IFN production and induction [45], and SARS-CoV-2 replication is dampened in Calu3 cells vs Vero cells due to this critical difference [46]. In neither of these cells, did the N40D mutant show much attenuation, as measured by quantification of viral particle production and intracellular viral RNA. In contrast, in primary human lung organoids and in K18-hACE2 mice, the N40D mutant was strongly attenuated and was associated with higher IFN induction. Our findings are reminiscent of what has been reported for other CoVs, and indicate that immortalized cells lines–even those like Calu3 deemed "IFN-competent" may not fully recapitulate innate immune responses during viral infection. Primary cells or *in vivo* models of infection are needed to fully characterize the functional role of certain viral proteins, including the Mac1 domain, in viral infection.

The N40D mutant virus yielded a dramatic survival phenotype in K18-hACE2 mice with all animals surviving while 100% of mice infected with the WT virus reached the humane endpoint. Infected K18-hACE2 mice usually die because of unnatural ACE2 overexpression on neurons and lethal encephalitis [41,47]. However, no infectious virus was recovered from the brains of animals infected with the N40D mutant, correlating with the attenuated viral titers recovered from the lungs of infected animals. It is possible that the Mac1 domain specifically supports neuronal replication of SARS-CoV-2 or that the IFN response when not effectively counterbalanced by the Mac1 domain's catalytic activity is vigorous in infected neurons similarly to the Mac1-deficient MHV-JHM (neurotropic) which induced higher levels of IFN in mice [30,31]. Interestingly, the N40D mutant replicated slightly higher than the WT virus at 2 days post-infection in the lungs of infected mice, which correlated with the lack of significant IFN induction at this timepoint. It is possible that elevated ADP-ribosylation of substrates outside the IFN response may support viral replication before the IFN response becomes effectively induced.

Despite the dramatic differences in survival and significantly faster viral clearance of the N40D mutant in mice, histology analysis of infected lungs demonstrated similar lung immune infiltration and pathology between the WT and N40D mutant virus. These data support the model that the N40D mutant virus can infect and replicate to robust levels early post-infection and therefore induces similar lung immune pathology, but is cleared quicker than the wild-type virus due to potentially attenuated host innate immune response antagonism. Our results are in contrast to a recently published SARS-CoV-2 NSP3 Mac1 deletion mutant where significantly less lung pathology was reported [34]. However, the deletion mutant appears to replicate significantly less at all timepoints post-infection compared with the wild-type virus suggesting additional defects in infectivity or replication that are not observed with the catalytic activity N40D mutant virus.

Future studies will identify the cellular targets of the Mac1 domain and the mechanism of its critical role in the viral lifecycle. Indeed, the importance of Mac1 in the CoVs lifecycle has led several groups to develop inhibitors of Mac1 catalytic activity by crystallographic screening [48,49] and high throughput *in vitro* screening [50–52]. However, the role of these compounds as potential SARS-CoV-2 therapeutics has not yet been tested in antiviral assays and will be investigated in future studies.

## Materials and methods

### Ethics statement

All research conducted in this study complies with all relevant ethical regulations. All experiments conducted with replication-competent viruses were performed in a certified biosafety level 3 (BSL3) laboratory and experiments were approved by the Institutional Biosafety Committee of the University of California, San Francisco and the Gladstone Institutes. All protocols

concerning animal use were approved (AN169239-01C) by the Institutional Animal Care and Use Committees of the University of California, San Francisco and the Gladstone Institutes and conducted in strict accordance with the National Institutes of Health Guide for the Care and Use of Laboratory Animals. All human samples utilized for generating human airway organoids were de-identified, not used to conduct human subject research, and were therefore IRB exempt.

## Cells

BHK-21 and A549 cells were obtained from ATCC and cultured in DMEM (Corning) supplemented with 10% fetal bovine serum (FBS) (GeminiBio), 1x glutamine (Corning), and 1x Penicillin-Streptomycin (Corning) at 37˚C, 5% $CO_2$. Calu3 cells were obtained from ATCC and cultured in Advanced DMEM (Gibco) supplemented with 2.5% FBS, 1x GlutaMax, and 1x Penicillin-Streptomycin at 37˚C and 5% $CO_2$. Vero cells stably overexpressing human TMPRSS2 (Vero TMPRSS2) (gifted from the Whelan lab [53]), were grown in DMEM with 10% FBS, 1x glutamine,1x Penicillin-Streptomycin at 37˚C and 5% $CO_2$. Vero cells stably co-expressing human ACE2 and TMPRSS2 (VAT) (gifted from A. Creanga and B. Graham at NIH) were maintained in DMEM supplemented with 10% FBS, 1x Penicillin-Streptomycin, and 10 μg/mL of puromycin at 37˚C and 5% $CO_2$.

## Plasmids

IDT gBlock gene fragments encoding WT and mutant SARS-CoV-2 Mac1 were cloned into a BamHI- and EcoRI-linearized pLVX-EF1alpha-nCoV2019-nsp13-2xStrep-IRES-Puro (Addgene, 141379) by Gibson Assembly Cloning reaction (NEB, 102715–912). Plasmids were validated by whole plasmid sequencing (Primordium). SARS-CoV-2 replicon and infectious clone plasmids were constructed as described previously [36]. Briefly, the N40A and N40D mutants were cloned by utilizing NEB HiFi DNA assembly kit onto the second fragment (W2) of the WA1 genetic background. The mutant plasmids were combined with the remaining nine fragments and assembled into the full BAC plasmid using Golden Gate assembly. The resulting plasmid was then maxiprepped and sequenced using Primordium Labs "Large" whole plasmid sequencing service. The Spike (S) and Nucleocapsid (N) expression vectors were described previously [54].

## SARS-CoV-2 replicon assay

The SARS-CoV-2 replicon assay was conducted as described previously [36]. Briefly, the pBAC SARS-CoV-2 Δ Spike WT, N40A, or N40D plasmid (1 μg), was transfected into BHK-21 cells along with N and S expression vectors (0.5 μg each) in 24-well. The supernatant was replaced with fresh growth medium 12–16 hours post-transfection. The supernatant containing single-round infectious particles was collected and 0.45 μm-filtered 72 hours post-transfection. The supernatant was subsequently used to infect VAT cells (in 96-well plate) or Calu3 cells (in 24-well plate). The medium was refreshed 12–24 hours post-infection. To measure luciferase activity, an equal volume of supernatant from infected cells was mixed with Nano-Glo luciferase assay buffer and substrate and analyzed on an Infinite M Plex plate reader (Tecan).

## Infectious clone virus rescue

pBAC SARS-CoV-2 WT (WA1) or N40D mutant constructs were directly cotransfected with an N expression vector into BHK-21 cells in 6-well plate. After 3 days, the supernatant was

collected and used to infect Vero TMPRSS2 cells and passaged further to achieve high viral titer. All viruses generated and/or utilized in this study were NGS verified using the ARTIC Network's protocol [55].

## SARS-CoV-2 virus culture and plaque assay

SARS-CoV-2 infectious clones were propagated on Vero TMPRSS2 cells, sequence verified, and were stored at -80˚C until use. For plaque assays, tissue homogenates and cell culture supernatants were analyzed for viral particle formation for *in vivo* and *in vitro* experiments, respectively. Briefly, VAT cells were plated and rested for at least 24 hours. Serial dilutions of inoculate of tissue homogenates or cell culture supernatants were added on to the cells. After the 1-hour absorption period, 2.5% Avicel (Dupont, RC-591) was overlaid. After 72 hours, the overlay was removed, the cells were fixed in 10% formalin for one hour, and stained with crystal violet for visualization of plaque formation.

## Real-time quantitative polymerase chain reaction (RT-qPCR)

RNA was extracted from cells, supernatants, or tissue homogenates using RNA-STAT-60 (AMSBIO, CS-110) and the Direct-Zol RNA Miniprep Kit (Zymo Research, R2052). RNA was then reverse-transcribed to cDNA with iScript cDNA Synthesis Kit (Bio-Rad, 1708890). qPCR reaction was performed with cDNA and SYBR Green Master Mix (Thermo Fisher Scientific) using the CFX384 Touch Real-Time PCR Detection System (Bio-Rad). The tenth fragment of the infectious clone plasmid [36] was used as a standard for N gene quantification by RT-qPCR. All primer sequences utilized in this study are included in S1 Table.

## K18-hACE2 mouse infection model

K18-hACE2 mice were obtained from the Jackson Laboratory and housed in a temperature- and humidity-controlled pathogen-free facility with 12-hour light/dark cycle and *ad libitum* access to water and standard laboratory rodent chow. Briefly, the study involved intranasal infection ($1X10^3$ PFU) of 6–8-week-old female K18-hACE2 mice with SARS-CoV-2 WA1 or N40D mutant virus. A total of 15 animals were infected for each virus and euthanized at 2, 4, and 6 days post-infection (5 mice per timepoint). The lungs and brains were processed for further analysis of virus replication.

## Histology

Mouse lung tissues were fixed in 4% PFA (Sigma Aldrich, Cat #47608) for 24 hours, washed three times with PBS and stored in 70% ethanol. All the stainings were performed at Histowiz, Inc (Brooklyn, NY), using the Leica Bond RX automated stainer (Leica Microsystems) using a Standard Operating Procedure and fully automated workflow. Samples were processed, embedded in paraffin, and sectioned at 4μm. The slides were dewaxed using xylene and alcohol-based dewaxing solutions. Epitope retrieval was performed by heat-induced epitope retrieval (HIER) of the formalin-fixed, paraffin-embedded tissue using citrate-based pH 6 solution (Leica Microsystems, AR9961) for 20 mins at 95˚C. The tissues were stained for H&E, dried, coverslipped (TissueTek-Prisma Coverslipper), and visualized using a Leica Aperio AT2 slide scanner (Leica Microsystems) at 40X. All slides were interpreted by a board-certified pathologist. Quantification of alveolar septal, bronchial wall, and vascular wall thickness were done using Aperio ImageScope.

## Cellular infection studies

Calu3 cells were seeded into 12-well plates. Cells were rested for at least 24 hours prior to infection. At the time of infection, medium containing viral inoculum was added on the cells. One hour after addition of inoculum, the medium was replaced with fresh medium. The supernatant and cells were harvested at 48 hours post-infection for downstream analysis.

## Human airway organoid culture

Human airway organoids were generated from human lung tissue as previously described [37]. Briefly, the lung tissues were obtained from the Matthay lab (UCSF) and underwent enzymatic digestion to obtain single cells, which were then resuspended in Basement Membrane Extract (BME, R&D Systems). This single-cell mixture was plated into droplets, which were submerged in a specialized human airway organoid (HAO) medium consisting of 1 mM HEPES (Corning), 1x GlutaMAX (Gibco), 1x Penicillin-Streptomycin (Corning), 10% R-spondin1 conditioned medium, 1% B27 (Gibco), 25 ng/mL noggin (Peprotech), 1.25 mM N-acetyl-cysteine (Sigma-Aldrich), 10 mM nicotinamide (Sigma-Aldrich), 5 nM heregulin-β1 (Peprotech), and 100 µg/mL Primocin (InvivoGen) in DMEM. This HAO medium was also supplemented with 5 µM Y-27632, 500 nM A83-01, 500 nM SB202190, 25 ng/mL FGF7, and 100 ng/mL FGF10 (all obtained from Stem Cell Technologies). The HAO medium was refreshed bi-weekly, and the droplets were transferred into a single-cell suspension every two weeks to facilitate growth. To induce differentiation, the HAO medium was substituted with a mixture of HAO medium and PneumaCult-ALI Medium (Stem Cell Technologies) at a 1:1 ratio.

## Human airway organoid infection

Differentiated organoids were plated in V-bottom plates (Greiner Bio-One) with each well containing 100,000 cells in HAO differentiation medium containing 2% BME. The organoids were then exposed to SARS-CoV-2 at MOI of 1. Following a 2-hour incubation with the inoculum, the supernatant was removed, and the cells were washed three times with PBS and fresh medium was added. The plates were incubated at 37˚C and 5% $CO_2$. After incubation, the supernatants were collected for plaque assays and HAOs were harvested for RT-qPCR analysis at 24, 48, and 72 hours post-infection.

## Mac1 expression and purification

SARS-CoV-2 NSP3 Mac1 ($P4_3$ construct, residues 3–169 [49]) was transformed into BL21 (DE3) *E. coli* cells (NEB, C2527H) and grown overnight at 37˚C lysogeny broth (LB) agar supplemented with carbenicillin (100 µg/mL). Expression starter cultures inoculated with a single colony (10 mL of LB supplemented with carbenicillin) were grown at 37˚C for eight hours and were then used to inoculate large scale cultures (1 L of autoinduction media ZYM-5052 [56]. Each 1 L culture was assembled using the following stocks: 960 mL ZY (10 g/L tryptone, 5 g/L yeast extract, prepared in water and autoclaved), 2 mL $MgSO_4$ (1 mM, prepared in water and autoclaved), 1 mL $FeCl_3$ (0.1 M, prepared in 100 mM HCl and sterile filtered), 20 mL 50xM (1.25 mM $Na_2HPO_4$, 1.25 mM $KH_2PO_4$, 2.5 mM $NH_4Cl$, 0.5 mM $Na_2SO_4$), 20 mL 50x5052 (250 g/L glycerol, 25 g/L glucose, 100 g/L lactose monohydrate, prepared in water and autoclaved). Cultures were grown at 37˚C until an optical density of 1, followed by growth at 20˚C for 14 hours. Cells were collected by centrifugation (4000 g, 15 minutes, 4˚C) and frozen at −80˚C. Mac1 was purified and the His-tag cleaved with Tobacco Etch Virus (TEV) protease following the published protocol [49]. The purified protein was frozen in liquid nitrogen and

stored at −80˚C. Point mutations to Asn40 were introduced using Gibson assembly [57] and proteins were purified similarly to WT, except that the N40A mutant was frozen at 10 mg/mL.

## PARP10 expression and purification

Human PARP10 (catalytic domain, residues 819–1007) with a TEV-cleavable His$_6$-tag was synthesized by Integrated DNA Technologies and cloned into a pET22b(+) vector using Gibson assembly. Protein was expressed similarly to Mac1, except the large-scale cultures were incubated at 18˚C for 14 hours. Cells were resuspended in Ni–nitrilotriacetic acid (NTA) binding buffer (50 mM HEPES (pH 7.5), 500 mM NaCl, 10 mM imidazole, 10% glycerol, and 0.5 mM Tris (2-carboxyethyl) phosphine (TCEP), 0.2% Triton X-100 supplemented with Turbo-Nuclease (5 U/mL; Sigma-Aldrich, T4330)) and lysed by sonication. Cell debris was collected by centrifugation (30,000 g, 30 minutes, 4˚C), and the lysate was applied to a 5-mL HisTrap HP column (Cytiva, 17524802). The column was washed with 25 mL of binding buffer followed by 25 mL of 5% Ni-NTA elution buffer (50 mM HEPES (pH 7.5), 500 mM NaCl, 500 mM imidazole, 10% glycerol, and 0.5 mM TCEP) and then the protein was eluted with 100% elution buffer. Eluted protein was diluted to 1.6 mg/mL using TEV reaction buffer (50 mM HEPES (pH 7.5), 150 mM NaCl, 1 mM dithiothreitol (DTT), and 5% glycerol) with recombinantly expressed TEV protease [58] added at a 1:20 mass:mass ratio (Mac1:TEV) giving a total volume of 30 mL. The TEV cleavage reaction was incubated at 4˚C for 16 hours in a 30 mL 10 kDa molecular weight cutoff dialysis cassette (ThermoFisher Scientific, 66830) dialyzing against 2 L TEV reaction buffer. Following TEV cleavage, the reaction was filtered (0.2 μm cellulose acetate), diluted to 60 mL using TEV reaction buffer and applied to a HisTrap HP column. The cleaved protein was washed off the column using 100% Ni-NTA binding buffer and further purified by size-exclusion chromatography using a HiLoad 16/600 Superdex 75 pg column (Cytiva, 28989333) equilibrated with SEC buffer (20 mM HEPES (pH 7.5), 300 mM NaCl, 10% glycerol and 0.5 mM TCEP). Protein concentration was determined by absorbance at 280 nm using an extinction coefficient of 13410 M-1 cm-1 calculated using the Protparam server (https://web.expasy.org/protparam/).

## PARP10 auto-MARylation and Western blotting

Auto-MARylation of PARP10 was carried with reference to the previously published protocol [51]. Purified PARP10 was diluted to 20 μM using MARylation buffer (SEC buffer supplemented with 0.02% Triton X-100 and 200 μM β-NAD+ (Roche, NAD100-RO)). The MARylation reaction was incubated at 37˚C for 1.5 hours, before filtering (0.2 μM cellulose acetate), concentration with a 10 kDa MWCO centrifugal concentrator (Amicon, UFC901024) and size-exclusion chromatography using a HiLoad 10/300 Superdex 75 pg column (Cytiva, 17517401) to separate the unreacted β-NAD+ (S8 Fig). Eluted fractions were concentrated to 4.2 mg/mL before being flash frozen in liquid nitrogen and stored at −80˚C. Auto-MARylation of PARP10 was assessed by Western blot using an anti-MAR/PAR-antibody (Cell Signaling, 83732). Aliquots of PARP10 taken before and after reaction with β-NAD+ (70 μM, 7.5 μg) were boiled for 5 minutes at 95˚C in SDS-PAGE loading dye, then separated by SDS-PAGE (Bio-Rad, 456–9036) and transferred to a nitrocellulose membrane (Bio-Rad, 1704158) using a Trans-Blot Turbo transfer system (Bio-Rad). The membrane was stained with Ponceau S (Cell Signaling, 59803S) before destaining with Tris buffered saline Tween-20 (TBST) and incubation for 1 hour with blocking buffer (TBST with 50 g/L blocking agent (Bio-Rad, 1706404)). The membrane was then incubated with primary antibody for 1 hour (1:1000 dilution in blocking buffer), washed with blocking buffer for 30 minutes, then incubated with secondary antibody for 1 hour (1:10000 dilution in blocking buffer, Cell Signaling, 7074S). The horse-

radish peroxidase conjugated to the secondary antibody was detected by incubating the membrane with 2 mL chemiluminescence substrate (Bio-Rad, 1705062) for five minutes and then imaged using a ChemiDoc XRS+ imager (Bio-Rad). To assess the removal of ADP-ribose by Mac1, MARylated PARP10 (70 μM, 7.5 μg) was incubated with Mac1 (20 μM, 2.2 μg) for 1 hour at room temperature. Reactions were stopped by boiling in SDS-PAGE loading dye and blotting for ADP-ribose as described above.

## NUDT5 expression and purification

Human NUDT5 (residues 1–219) was expressed from a pET21b vector containing a 3C-protease cleavage N-terminal His$_6$ tag [59] using BL21(DE3) *E. coli* cells co-transformed with a pKJE7 plasmid expressing the chaperones DnaK, DnaJ and GrpE (Takara, 3340). Cells were grown overnight at 37˚C on lysogeny broth (LB) agar supplemented with carbenicillin (100 μg/mL) and chloramphenicol (25 μg/mL). NUDT5 was expressed using autoinduction media with the same method as PARP10, except that media was supplemented with chloramphenicol (25 μg/mL) and chaperone expression was induced with 0.05% arabinose (5 mL of a 100 g/L stock), added when the cultures were cooled from 37˚C to 18˚C. To purify NUDT5, cells were resuspended in Ni-NTA binding buffer (50 mM HEPES (pH 7.5), 300 mM NaCl, 10 mM imidazole, 5% glycerol, and 0.5 mM TCEP supplemented with TurboNuclease (5 U/mL)) and lysed by sonication. Cell debris was collected by centrifugation, and the lysate was applied to a 5-mL HisTrap HP column. The column was washed with 25 mL of binding buffer followed by 25 mL of 5% Ni-NTA elution buffer (50 mM HEPES (pH 7.5), 300 mM NaCl, 300 mM imidazole, 5% glycerol, and 0.5 mM TCEP) and then eluted with 100% elution buffer. The wash and elution fractions were combined and concentrated to 12 mL using a 10 kDa centrifugal concentrator and desalted into 3C reaction buffer (150 mM NaCl, 50 mM HEPES (pH 7.5), 5% glycerol and 0.5 mM TCEP) using a HiPrep 26/10 desalting column. The sample was diluted to 1.5 mg/mL, and PreScission protease (Cytiva, 27084301) was added to a concentration of 20 U/mg NUDT5 and the reaction was incubated at 4˚C overnight. The sample was further purified by anion exchange (AEX) chromatography using a HiTrap Q HP column (Cytiva, 17115401). First, the salt concentration of the sample was reduced to 100 mM using AEX no-salt buffer (50 mM HEPES (pH 7.5), 5% glycerol, 0.5 mM TCEP). Next, the sample was loaded onto the AEX column pre-equilibrated with AEX low-salt buffer (50 mM HEPES (pH 7.5), 100 mM NaCl, 5% glycerol, 0.5 mM TCEP). Bound protein was eluted with a gradient of 0-to-100% high-salt buffer (50 mM HEPES (pH 7.5), 1 M NaCl, 5% glycerol, 0.5 mM TCEP) over 75 mL. The flow through was concentrated and subjected to size-exclusion chromatography using a HiLoad 16/600 Superdex 200 pg column (Cytiva, 28989335) equilibrated with SEC buffer (20 mM HEPES (pH 7.5), 150 mM NaCl and 5% glycerol) (S9 Fig). Eluted fractions were concentrated to 4.3 mg/mL before being flash frozen in liquid nitrogen and stored at −80˚C. The purified NUDT5 was analyzed by SDS-PAGE and the mass confirmed by LC-MS (S9 Fig) using a Waters Acquity LC connected to a Waters TQ detector with electrospray ionization. Briefly, NUDT5 was diluted to 10 μM using 150 mM NaCl and 20 mM HEPES (pH 7.5) and then 5 μL was injected onto a C4 column held at 40˚C. The sample was separated with an initial wash of 95% solvent A (water with 0.1% formic acid) and 5% solvent B (acetonitrile with 0.1% formic acid) for 1.5 min, followed by a linear gradient to 95% solvent B over 6.5 minutes, and a final wash with 95% solvent B for 2 min. All steps were run at 0.2 mL/min.

## PARP10 solution assay

The ADP-ribosyl hydrolase activity of macrodomains was determined using the previously described method [59] with auto-MARylated PARP10 as substrate [51]. ADP-ribose produced

by Mac1 is hydrolyzed by the NUDT5 phosphodiesterase to AMP, which is detected using the AMP-Glo assay kit (Promega, V5011). The activity of the recombinantly expressed NUDT5 with ADP-ribose was determined by serially diluting NUDT5 from 2 to 0.002 μM using PARP10 reaction buffer (20 mM HEPES pH 7.5, 100 mM NaCl, 10 mM MgCl$_2$, 0.5 mM TCEP). NUDT5 (4 μL, 1 to 0.001 μM final concentration) was added to a white 384-well plate (Corning, 3824), and the reaction was initiated with 40 μM ADP-ribose (4 μL, 20 μM final concentration) (Sigma-Aldrich, A0752). After incubation at room temperature for 1 hour, the AMP concentration was quantified by incubation with AMP-Glo reagent I (8 μL) for 1 hour, followed by incubation with a 1:100 mixture of AMP-Glo reagent II and Kinase-Glo one solution (16 μL) for 1 hour. The plate was spun for 1 minute to remove bubbles, and luminescence was measured using a BioTek Synergy HTX plate reader. Values were corrected by subtracting a no-NUDT5 control and plotted as a function of NUDT5 concentration (S8 Fig). To quantify the amount of ADP-ribose generated by Mac1, a standard curve was prepared by serially diluting ADP-ribose from 20 to 0.4 μM using PARP10 reaction buffer. ADP-ribose (4 μL, 10 to 0.2 μM final concentration) was added to a 384-well plate and the reaction was initiated with 200 nM NUDT5 (4 μL, 100 nM final concentration) prepared using PARP10 reaction buffer. After incubation at room temperature for 1 hour, the concentration of AMP was determined as above. The plot of luminescence as a function of ADP-ribose concentration was non-linear at low concentrations of ADP-ribose, therefore the standard curve was fit with a four-parameter logistic equation by non-linear regression using GraphPad prism (S8 Fig).

Next, Mac1 and MARylated PARP10 were titrated to determine the MARylated PARP10 concentration and a suitable concentration of Mac1 to use in activity assays. Although the PARP10 concentration was estimated from absorbance at 280 nM, the concentration of ADP-ribose groups attached to PARP10 can be quantified by measuring the amount of ADP-ribose released upon treatment with excess Mac1. Wild type Mac1 was serially diluted from 4 to 0.004 μM and MARylated PARP10 was serially diluted from 40 to 2.5 μM, both using PARP10 reaction buffer. Mac1 (2 μL, 1 to 0.001 μM final concentration) was added to a 384-well plate along with 400 nM NUDT5 (2 μL, 100 nM final concentration) and the reaction was initiated with the serially diluted PARP10 (4 μL, 20 to 1.25 μM final concentration). After incubation at room temperature for 1 hour, the concentration of AMP was determined as above. Substrate depletion was observed at enzyme concentrations ~100 nM and higher, therefore a Mac1 concentration of 5 nM was selected to ensure that the single measurements of ADP-ribose concentration were taken when product formation was linear with respect to time (e.g. when less than 20% of the total substrate was consumed). The concentration of MARylated PARP10 was determined by converting luminescence values at the highest concentration of Mac1 (1 μM) into ADP-ribose concentration using the standard curve (S8 Fig). At this Mac1 concentration, all PARP-10 bound ADP-ribose is hydrolyzed to free ADP-ribose.

To determine the rate of Mac1 catalyzed hydrolysis of MARylated PARP10, a mixture of 10 nM Mac1 and 200 nM NUDT5 (4 μL, final concentration of 5 nM for Mac1 and 100 nM for NUDT5) was added to a 384-well plate. The reaction was initiated by adding serially diluted MARylated PARP10 (4 μL, 3.6 to 0.22 μM final corrected concentration) and the reaction was incubated at room temperature for 1 hour followed by AMP detection as above. Enzyme velocities in RLU/min were converted to nM ADP-ribose/min using the standard curve. The increase in enzyme velocity was linear with respect to substrate concentration, suggesting that the [substrate] $<< K_M$, which is consistent with the previously determined $K_M$ for Mac1 with a PARP10 derived peptide (163 μM). The catalytic efficiency of Mac1 ($k_{cat}/K_M$) was calculated dividing the slope of the substrate-velocity curve by the enzyme concentration, because at substrate concentration $<< K_M$ the hydrolysis reaction is first order with respect to both substrate and enzyme concentration. The activity of the N40D/A mutants with auto-MARylated

PARP10 were measured as above, however the enzyme concentrations were adjusted to give similar velocities to 5 nM Mac1 (N40D = 500 nM, N40A = 50 nM).

## X-ray crystallography

Crystals of the N40D mutant in the $P4_3$ space group were grown at 292 K by microseeding with wild-type $P4_3$ crystals using sitting-drop vapor diffusion as described previously [49]. The reservoir solution contained 28% PEG 3000 and 100 mM CHES (pH 9.5). Crystals grew overnight and were vitrified directly in liquid nitrogen without additional cryoprotection. The structure with ADP-ribose bound was obtained by soaking a crystal in a solution of 20 mM ADP-ribose prepared in 32% PEG 3000 and 100 mM CHES (pH 9.5) for two hours at room temperature prior to vitrification in liquid nitrogen. X-ray diffraction data were collected at 100 K using beamline 8.3.1 of the Advanced Light Source. Data collection strategy and statistics are shown in S2 Table.

Phases were obtained by molecular replacement with *Phaser* [60] using a model of apo Mac1 derived from PDB code 7KQO with coordinates randomly shifted by 0.3 Å to remove model bias from the apo structure [61]. Refinement of the apo model was performed with *phenix.refine* [62] using default settings and five macrocycles at each step. Water molecules were added automatically in phenix.refine to peaks in the $mF_O–DF_C$ map 3.5 σ or higher. Atomic displacement parameters (ADPs) were refined isotropically in the first two cycles of refinement and anisotropically in the remaining cycles. After three cycles of refinement and model building with *Coot* [63], hydrogens were added with *Reduce* [64] run through *phenix.ready_set*. Hydrogens were refined using a riding model, with their ADPs refined isotropically. The occupancy of all water molecules was refined in the final two stages of refinement. For the dataset obtained from a crystal soaked with 20 mM ADP-ribose, ligand binding was confirmed by inspection of $mF_O–DF_C$ electron density maps, with restraints generated by *phenix.elbow* [65]. Refinement statistics are shown in S2 Table.

## Differential scanning fluorimetry (DSF)

DSF experiments were performed according to the previously described protocol [49]. Briefly, ADP-ribose was dissolved in $H_2O$ to a final concentration of 200 mM, and serial 1:2 dilutions created a ten-point concentration series from 200 to 0.39 mM. DSF buffer was prepared by diluting SYRPO orange (Thermo Fisher Scientific, S6650) from 5000x to 10x in 50 mM Tris (pH 7.5), 150 mM NaCl, 1 mM EDTA, 1 mM DTT, 0.01% Triton X-100. The ADP-ribose concentration series was then diluted 1:100 with DSF buffer (2 μL + 198 μL DSF buffer) and 10 μL was aliquoted into a 384-well white PCR plate (Axygen, PCR-384-LC480-W-NF) and the plates were incubated in the dark for 20 minutes. Purified Mac1 or buffer was then added to each well (10 μL at 4 μM). Each well had a total of 20 μL that contained 2 μM protein (or buffer only), 5x SYPRO orange and 1 to 0.002 mM ADP-ribose (or water only). Each condition was repeated four times, while the no-ADP-ribose conditions were repeated eight times. The PCR plate was sealed with a Microseal 'B' seal (Bio-Rad, MSB1001) and spun to remove air bubbles. The temperature ramp was performed using a BioRad CFX 384 qPCR instrument with fluorescence monitored using the FRET channel from 25 to 95˚C at a rate of 1˚C/min. Plots of raw RFU values are shown in S10 Fig. $T_{ma}$ values were calculated online using DSFworld using fitting model 1 [66].

## Generation of A549 Mac1-WT and mutant cell lines

Lentivirus was generated using standard protocols and supernatants were collected at 48 hours post-transfection, filtered through a 0.45 μm membrane and added to A549 cells 1:1 in

complete media supplemented with 8 μg/mL polybrene. The media was changed after 24 hours and, after 48 hours, media containing 2 μg/mL puromycin was added. Cells were selected for 48 hours and then expanded without further selection.

## Western blotting for SARS-CoV-2 Mac1

Cell lysates were prepared in Pierce RIPA Buffer (Thermo Fisher Scientific, 89901) containing protease inhibitor and PhosStop cocktails (Roche, 5892970001 and 4906845001, respectively), separated on an SDS-polyacrylamide gel and transferred to a PVDF membrane and probed. The following primary antibodies and dilutions were used: Strep Tag (Thermo Scientific, MA517282, 1:1000) and β-actin-HRP (Cell Signaling, 5125, 1:2000). Images were captured using the Azure c600 Western Blot Imaging System.

## Supporting information

**S1 Fig. Alignment of Mac1 sequences from several betacoronaviruses.** The Mac1 sequences of SARS-CoV-2, SARS-CoV, MERS-CoV, and MHV-A59 were obtained from uniport (P0DTC1, P0C6U8, K9N638, and P0C6V0, respectively) and aligned with Geneious Prime version 2022.2.1 using the Clustal Omega setting. Residues shaded in black are identical across all viruses, while residues shaded in grey are only partially conserved. The asparagine at position 40 that is critical for catalytic activity is indicated in a red rectangle.
(TIF)

**S2 Fig. Mac1 N40A protein is less stable in cells than are Mac1 WT or Mac1 N40D proteins.** Western blot analyses of cell lysates from A549 lung cancer cells lentivirally transduced with vectors expressing Strep-tagged Mac1 WT, Mac1 N40D or Mac1 N40A proteins. β-actin is shown as a loading control. The blot is representative of three independent replicates.
(TIF)

**S3 Fig. Structure of Mac1 N40D determined by X-ray crystallography in apo- and ADP-ribose-bound state.** (**A**) Alignment of apo- (PDB code 8SH6) and ADP-ribose-bound N40D (PDB code 8SH8) structures shows previously identified structural changes occurring upon ADP-ribose binding (e.g. Ala129 flip, Phe132 and Asn99 rotation, shown with orange arrows) [49]. Selected hydrogen bonds between ADP-ribose and the active site are shown with dashed black lines. Three water molecules are shown that form bridging interactions between ADP-ribose and active site residues (purple/blue spheres). (**B**) Alignment of apo- WT (PDB code 7KQO) and N40D structures shows that there are only minor structural changes in the active site. The 80° rotation of the aspartic acid side chain relative to the asparagine side chain is indicated with a black arrow. (**C**) Alignment of ADP-ribose-bound WT (PDB code 7KQP) and N40D (PDB code 8SH8) shows that ADP-ribose binding is highly conserved despite the N40D mutation (RMSD across the 36 ADP-ribose atoms of 0.12 Å). The bridging water that is unique to N40D is shown with a purple sphere. Two conserved bridging waters are shown with blue spheres.
(TIF)

**S4 Fig. IFNβ treatment does not differentially suppress the replication of the SARS-CoV-2 replicon with Mac1 N40D mutation.** Calu3 cells were infected with indicated replicons in the presence of indicated IFNβ concentrations. Intracellular viral RNA (N gene copies) was measured by RT-qPCR using a standard curve. Data are presented as mean +/- SD of one biological replicate conducted in triplicate.
(TIF)

**S5 Fig. SARS-CoV-2 Mac1 domain N40D mutant replicates similarly compared with WT virus regardless of infectious dose.** Viral particle release of infected Calu3 cells at MOI 0.01 and MOI 0.1 with indicated viruses was measured at indicated timepoints post-infection by plaque assay on VAT cells. Data are presented as mean +/- SD of three biological replicates conducted in duplicate. **, $p < 0.01$ by two-sided Student's T test.
(TIF)

**S6 Fig. SARS-CoV-2 Mac1 N40D mutant is attenuated in K18-hACE2 mice.** A) Viral particle abundance in the upper respiratory tract of mice infected with WT or Mac1 N40D mutant viruses was measured by plaque assay in VAT cells. Data are presented as mean +/- SD. B) Viral RNA levels (N gene copies) in the upper respiratory tract were measured by RT-qPCR using a standard curve. Data are presented as mean +/- SD. C) Viral particle abundance in the brains of mice infected with WT or Mac1 N40D mutant viruses at 6 days post-infection was measured by plaque assay in VAT cells. Data are presented as mean +/- SD. *, $p < 0.05$; **, $p < 0.01$ by two-sided Student's T-test.
(TIF)

**S7 Fig. SARS-CoV-2 Mac1 N40D mutant has similar pathogenesis in K18-hACE2 mice to the wild-type virus.** Four pathology scoring criteria (A-D) were utilized to assess the pathogenesis of the wild-type and mutant viruses in K18-hACE2 mice. The total score encompassing these criteria is in Fig 4H. The extent of inflammation in the perivascular (E), peribronchial (F), and interstitial (G) areas was quantified by measuring the vascular wall, the bronchial wall, and the alveolar septal thickness, respectively. Data are presented as mean +/- SD. *, $p < 0.05$; **, $p < 0.01$; ***, $p < 0.001$; ****, $p < 0.0001$ by two-sided Student's T-test.
(TIF)

**S8 Fig. Purification and quantification of MARylated PARP10.** (**A**) Size exclusion chromatography (HiLoad 10/300 Superdex 75 pg column) of PARP10 with and without treatment with β-NAD+. The peaks eluting at ~18 mL are un-reacted β-NAD+ and/or hydrolyzed ADP-ribose. (**B**) SDS-PAGE (Biorad, 4569036) analysis of MARylated PARP10 with protein marker (ThermoFisher, 26616). Samples were boiled in SDS-PAGE loading dye for five minutes at 95˚C and the gel was stained with Coomassie blue. (**C**) Titration of human NUDT5 phospho-diesterase with 20 μM ADP-ribose. AMP was detected using the AMP-Glo kit. Each condition was measured three times. (**D**) Log-log plot of the titration of ADP-ribose with 100 nM NUDT5. The highest three concentrations of ADP-ribose were fit using linear regression with GraphPad Prism (red line) and the remaining concentrations were fit to a four-parameter logistic equation using non-linear regression (gray line). Each condition was measured three times. (**E**) Titration of MARylated PARP10 and Mac1. Each plot shows the titration of Mac1 with a single concentration of MARylated PARP10. Each condition was measured twice, with the maximum luminescence determined by fitting a four-parameter logistic equation using non-linear regression (gray line). (**F**) ADP-ribose concentration generated by MARylated PARP10 hydrolysis plotted as a function of MARylated PARP10 concentration determined by absorbance at 280 nm. Data were fit by linear regression (gray line), with the line constrained to the X = 0, Y = 0 (the slope and standard error are indicated).
(TIF)

**S9 Fig. Purification of human NUDT5.** (**A**) Size exclusion chromatography (HiLoad 16/600 Superdex 200 pg column) of NUDT5 in the final purification step. The monomeric fractions pooled are shaded orange. (**B**) SDS-PAGE analysis of purified NUDT5 with protein marker (ThermoFisher, 26616). Samples were boiled in SDS-PAGE loading dye for five minutes at 95˚C and the gel was stained with Coomassie blue. (**C**) Mass spectrum of purified NUDT5

deconvoluted with the maximum entropy method (MaxEnt1). The observed mass is within 3 Da of the mass calculated from the amino acid sequence (24619 Da).
(TIF)

**S10 Fig. Thermostability of WT, N40D and N40A mutants assessed by differential scanning fluorimetry with SYPRO orange.** (**A**) Plot showing apparent melting temperatures ($T_{ma}$) for WT, N40D and N40A Mac1 as a function of ADP-ribose concentration. $T_{ma}$ values were calculated by DSFworld [66] using fitting model 1. Data are presented as the mean ± standard deviation of four technical replicates. (**B**) Raw fluorescence curves used to calculate $T_{ma}$ values. Data are presented as the mean of four technical replicates.
(TIF)

**S1 Table. RT-qPCR primers utilized in this study.**
(DOCX)

**S2 Table. Data collection and refinement statistics for X-ray crystal structures reported in this work.**
(DOCX)

## Acknowledgments

We thank the Whelan laboratory for providing the Vero cells overexpressing human TMPRSS2 and A. Creanga and B. Graham for the Vero cells overexpressing human ACE2 and TMPRSS2. We thank the Gladstone Histology and Light Microscopy Core and Dr. Blaise Ndjamen for assistance with the mice lung pathology analysis. We thank M. Matthay lab for providing lung tissues for generating human airway organoids. We thank Francoise Chanut for feedback on the manuscript. The synchrotron X-ray diffraction data used to determine Mac1 structures were collected at beamline 8.3.1 of the Advanced Light Source (ALS). The ALS, a U.S. DOE Office of Science User Facility under contract no. DE-AC02-05CH11231, is supported in part by the ALS-ENABLE program funded by the NIH, National Institute of General Medical Sciences, grant P30 GM124169-01.

## Author Contributions

**Conceptualization:** Taha Y. Taha, Melanie Ott.

**Data curation:** Taha Y. Taha, Rahul K. Suryawanshi, Irene P. Chen, Galen J. Correy, Maria McCavitt-Malvido, Patrick C. O'Leary.

**Formal analysis:** Taha Y. Taha, Rahul K. Suryawanshi, Irene P. Chen, Galen J. Correy, Maria McCavitt-Malvido.

**Funding acquisition:** Irene P. Chen, Nevan J. Krogan, Alan Ashworth, James S. Fraser, Melanie Ott.

**Investigation:** Taha Y. Taha, Rahul K. Suryawanshi, Irene P. Chen, Galen J. Correy, Maria McCavitt-Malvido, Patrick C. O'Leary, Manasi P. Jogalekar, Morgan E. Diolaiti, Gabriella R. Kimmerly, Chia-Lin Tsou, Ronnie Gascon, Mauricio Montano.

**Methodology:** Taha Y. Taha, Rahul K. Suryawanshi, Irene P. Chen, Galen J. Correy, Maria McCavitt-Malvido, Patrick C. O'Leary, Manasi P. Jogalekar, Morgan E. Diolaiti, Gabriella R. Kimmerly, Chia-Lin Tsou, Ronnie Gascon, Mauricio Montano.

**Project administration:** Alan Ashworth, James S. Fraser, Melanie Ott.

**Resources:** Alan Ashworth, James S. Fraser, Melanie Ott.

**Supervision:** Alan Ashworth, James S. Fraser, Melanie Ott.

**Validation:** Taha Y. Taha, Rahul K. Suryawanshi, Irene P. Chen, Galen J. Correy, Maria McCavitt-Malvido.

**Visualization:** Taha Y. Taha, Rahul K. Suryawanshi, Irene P. Chen, Galen J. Correy, Maria McCavitt-Malvido.

**Writing – original draft:** Taha Y. Taha, Irene P. Chen, Galen J. Correy, Melanie Ott.

**Writing – review & editing:** Taha Y. Taha, Irene P. Chen, Galen J. Correy, Maria McCavitt-Malvido, Morgan E. Diolaiti, Luis Martinez-Sobrido, Alan Ashworth, James S. Fraser, Melanie Ott.

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
