## [Decision Letter · Decision Letter 0]

12 Jun 2023

Dear Dr. Ott,

Thank you very much for submitting your manuscript "A single inactivating amino acid change in the SARS-CoV-2 NSP3 Mac1 domain attenuates viral replication and pathogenesis in vivo" for consideration at PLOS Pathogens. As with all papers reviewed by the journal, your manuscript was reviewed by members of the editorial board and by several independent reviewers. In light of the reviews (below this email), we would like to invite the resubmission of a significantly-revised version that takes into account the reviewers' comments.

We cannot make any decision about publication until we have seen the revised manuscript and your response to the reviewers' comments. Your revised manuscript is also likely to be sent to reviewers for further evaluation.

Sincerely,

Jacob S. Yount

Academic Editor

PLOS Pathogens

Sonja Best

Section Editor

PLOS Pathogens

Kasturi Haldar

Editor-in-Chief

PLOS Pathogens

orcid.org/0000-0001-5065-158X

Michael Malim

Editor-in-Chief

PLOS Pathogens

orcid.org/0000-0002-7699-2064

Reviewer's Responses to Questions

**Part I - Summary**

Reviewer #1: The manuscript "A single inactivating amino acid change in the SARS-CoV-2 NSP3 Mac1 domain attenuates viral replication and pathogenesis in vivo" by Tara et al described the characteristics of a loss-of-function mutation in NSP3 Mac1 domain of SARS-CoV-2 by using both in vitro and in vivo infection model. The study added some new facets of Mac1 domain in virus replication and anti-viral responses modification. However, the inappropriate description and interpretation of data diminished the convince of their findings.

Reviewer #2: In the manuscript entitled “A single inactivating amino acid change in the SARS-CoV-2 NSP3 Mac1 domain attenuates viral replication and pathogenesis in vivo” Taha, Suryawanshi, Chen, Correy et al examine the effects of inactivating point mutations in the macrodomain1 in NSP3 on viral replication and host responses to infection. Overall, they demonstrate that inactivation of N40 by substitution with alanine residues renders NSP3 unstable. On the other hand, mutation of N90 to aspartic acid ablated catalytic activity without affecting protein stability. The authors then implemented single-round replicon to demonstrate that NSP3 N90D induces increased host antiviral responses relative to the WT virus. The authors then employed infectious clones to demonstrate that N90D had decreased modestly in Calu3 and human airway organoids. Lastly, in vivo studies demonstrate that the N90D attenuates the pathogenicity of SARS-CoV-2 relative to WA-1. These studies suggest that Mac1 of NSP3 is a virulence determinant as has been previously shown in other coronaviruses. Mechanistically, the authors suggest that this is due to an inability to counter the host antiviral response. This study is novel in demonstrating the relevance of the catalytic activity of AC1 in NSP3. However, the manuscript in its current form is too preliminary to support the hypothesis that antagonism of host IFN responses is the mechanism by which Mac1 contributes to SARS-CoV2 viral virulence rather than direct effects on viral replication.

Reviewer #3: The manuscript by Taha et al provides characterisation of the novel catalytic mutant of the coronavirus Mac1 domain enzyme (N40D) that is superior to the previously utilised N40A mutant. Importantly, the N40D virus replicates 1000 times less efficiently compared to the wild-type virus in mice. This mutant also induces an efficient interferon response. The presented data validate the SARS-CoV2 NSP3 Mac1 domain as a critical viral pathogenesis factor and a promising target to develop antiviral drugs. Altogether, this is an important, well-written and timely paper.

Reviewer #4: From the Introduction to the manuscript, SARS-CoV-2 NSP3 Mac1 has been shown to be “necessary for pathogenesis and for robust viral replication” (references cited in Introduction by authors 28-31; quotes are wording used by authors). Mac1 has been shown to be an ADP-ribosylhydrolase that can remove ADP-ribosylation and hence decrease ADP-ribosylation-dependent events/pathways. Mac1 has a conserved asparagine at position 40 in the active site. Mutation of asparagine to alanine results in inactivation of Mac1. Mac1 is found in the SARS-CoV family of viruses. Immortalized cell lines may not recapitulate the effects of Mac1. Deletion of Mac1 did not appear to affect in vitro activity but did affect in vivo activity. Mac1 deletion mutant was associated with reduced effects in vivo.

The investigators in this report focus on SARS-CoV-2 and show that Mac1 is critical in SARS-CoV-2 for viral replication and pathogenesis in vivo. They compare substitutions at the critical asparagine with either alanine or asparagine. Both substitutions decrease enzymatic activity of Mac1, with focus of the N40D mutant. The studies on N40D are detailed and well done, in terms of choice of model systems and structural analysis of the mutant Mac1. Different organ responses were observed, for reasons that are not addressed experimentally. The novelty of the report is more related to the detailed analysis of in vivo, in vitro and structure analyses.

It is not clear if the results change prior literature on the importance of Mac1 to pathogenesis of disease, and by inference Mac1 enzymatic activity.

**Part II – Major Issues: Key Experiments Required for Acceptance**

Reviewer #1: Major concerns:

1 The authors claimed that "The N40D mutant generated similar luciferase signals to the WT replicon in both VAT and Calu3 cells... (Fig. 2B).", as well as "This mutation, when introduced via reverse genetics into fully infectious viral clone or replicons, showed minimal effects on viral replication in immortalized cell lines while modestly inducing the IFN response". However, in Figure 2B, the authors showed significant difference in RLU signals between WA1 and WA1 Mac1 N40D (p<0.001), which can not be treated as "similar" or "mininal" and may affect the conclusion drawn from these data. In addition, the authors are strongly recommended to repeat these assays by using rescued virus carrying point mutations (especially N40D).

2 The authors need to be very careful and specific in wording "IFN signal". IFN signals can be comprised of diverse pathways demonstrating distinct functions during SARS-CoV-2 infection and should not be treated equally. In Fig. 2D, the authors showed that WA1 Mac1 N40D infection induced higher IFNb1 but did not affect the expression of STAT1, it is confusing that in subsequnt assays (Fig. 2E), the authors used IFN-gamma (more stat1-related) instead of IFN-beta to treat infection system. These assays need to be justified.

3 The authors claimed that "at 48 hours after infection, particle production was measured by plaque assays while intracellular viral RNA and cytokine expression levels were measured by RT-qPCR. The N40D Mac1 mutant showed modestly (~2-fold) lower particle production and intracellular viral RNA compared with the WT virus suggesting a slightly attenuated phenotype (Fig. 2C and 2D). Similar to the results from replicon experiments, the N40D mutant induced higher levels of IFN as compared to WT virus (Fig. 3D)", but did not showed any statistics outcome in corresponding figures.

4 The authors interpreted data shown in Fig. 3F as "loss of the IFN antagonism

limits viral infection by the mutant virus, especially at later time points after infection, and differs from previous work with SARS-CoV where the Mac1-deficient virus phenotype is apparent early post infection". The explanation is very confusing. In addition, the decrease of virus titer at 48 and 72 hours could not be attributed to less efficient infection. Instead, it showed the accelerated virus clearance to me.

5 The authors need to have independent pathologists check and score the lung pathogenic changes shown in Fig. 4G.

Reviewer #2: The in vitro data in its current state does not robustly support claims that the host antiviral response is the mechanism for the attenuation of N90D. Loss of NSP3 Mac 1 catalytic activity could be impacting viral replication directly.

1. Please clarify whether equivalent infectious doses used to infect cells with replicon and recombinant viruses. What is the proportion of cells that is infected/transduced by WT and N40D viruses? Does decreasing the infectious dose enhance the detection of the replication defect of the catalytic mutant?

2. In figure 2E the authors treat cells with IFN gamma to determine whether the cell intrinsic IFN induction could suppress the N40D mutant. IFN gamma is not synthesized by infected Calu3 cells, rather viral infection induces the expression of type I and/or type III IFNs. Thus, it would be more appropriate to contrast cell intrinsic IFN sensitivity to conclude that Mac1 is required to overcome antiviral immunity. Furthermore, the authors conclude that IFN gamma does not hinder WT infection at either 10or100ng/ml concentration, yet there is a two-log decrease N copy number. Have statistics been run on these data? Is this change significant given that there is only a ~2 fold change between mutants across doses? While this is cited in the figure legends, there is no indication of significance on the figure itself. If so, a Student’s t test is not an appropriate statistic for these comparisons (effect mutants and effects doses). A better way to determine whether the host antiviral response contributes to viral attenuation of N90D would be to infect IRF3-deficient cells and determine whether the 2-fold attenuation is conserved.

3. As the overall hypothesis of this study is that the host innate immune response is most competent in primary tissues, it is important to demonstrate that decreases in viral titers of Mac1 mutants is also concordant with increased antiviral and inflammatory responses in human organoids (Fig 3F). Are these changes more robust than the modest changes in gene expression seen in Calu3 cells? As the authors conclude that “It is possible that the modest phenotype of the N40D mutant observed in Calu3 cells is due to these cells not fully recapitulating the cellular environment during SARS-CoV-2 infection” and go on to further expand upon this concept in the discussion, the production of infectious virus should be measured at 72hrs post infection. The differences in infectious virus production in Calu-3 were similar to those observed at 48hrs in organoids. The sample size should also be increased in order to determine whether any of the changes in infectious virus production or gene expression are significant between viruses.

Reviewer #3: None.

Reviewer #4: No major issues noted on the conduct of the studies in the analysis of the role of N40D. The data are more relevant to basic understanding of Mac1 then to approaches for therapy. The investigators might want to change the Discussion to address this point.

**Part III – Minor Issues: Editorial and Data Presentation Modifications**

Reviewer #1: Minor concerns

The authors need to double-check the typos and figure indicators.

In "The N40D Mac1 mutant showed modestly (~2-fold) lower particle production and intracellular viral RNA compared with the WT virus suggesting a slightly attenuated phenotype (Fig. 2C and 2D).", the correlating figures should be 3B and 3C.

In "While WT and Mac1 N40D mutant viruses produced similar levels of infectious particles at 24 hours post infection, the Mac1 N40D mutant showed ~10-fold lower particle production at 48 and 72 hours post infection (Fig. 4B).", the correlating figures should be 3F.

Finally, the authors are suggested to re-write or divide some long sentences and make the points clearer.

Reviewer #2: Additional editing of the manuscript and statistical analysis is necessary to better understand experimental design and data interpretation.

1. The authors conclude in Figure 2B that the relative luciferase signals of WT and N90D mutant are similar, yet the statistical analysis suggest differences in the RLU between these two groups. The authors should clarify whether a modest decrease in infectious virus packaging was observed.

2. The authors identify IFNB1, IL-6, STAT1 and ISG15 as cytokines. Only IFNB1 and IL6 fit that description and STAT1 and ISG15 are best described as ISGs. Similarly in the date presented in Fig 4. MX1 is not a cytokine. Consider updating this.

3. Could the authors clarify the methodology for quantifying the relative expression of host innate immune genes across figures 2and 3? Are the changes in gene expression conserved when contrasting to a house keeping gene rather than a spike control? There could be differential cell death across mutants resulting in changes in gene detection.

4. The authors should revise the figure callouts and legends. Figure legend labels for Figure 4G and 4H are swapped. Refer to figures 3B and 3C in main text. Figure 3F is called out as figure 4B in main text.

5. Could the authors please clarify what is mean by the statement: “underscoring the specific effect of this mutation in the viral life cycle.” Is the specific effect innate immune activation?

6. How many animals does the data presented in figure correspond to for each infectious group (days post infection)? 5 each?

7. For figure 4G consider highlighting the area of the lung that was magnified and including images from replicate lungs. Quantification of lung pathology would increase rigor. Does N90D infect areas of the lung distinct from those targeted by WA-1?

Reviewer #3: Minor comments:

Introduction – ‘Specifically, PARP7, PARP9, PARP12 and PARP14 directly enhance antiviral innate immune responses (24-26) and are upregulated during MHV infection (27).’ – I don’t think PARP7 enhances antiviral responses, it’s quite the opposite.

Introduction – it would be very good to clearly mention that PARP14 is the main antiviral PARP acting against coronaviruses and that Mac1 reverses PARP14-mediated ADP-ribosylation (doi: 10.1371/journal.ppat.1007756; doi: 10.1098/rsob.200237).

Supplementary Figure 1 – it would be good to label the catalytic residue in the figure.

Discussion – considering that the presented results greatly motivate further efforts in development of Mac1 inhibitors, it would be good to discuss more extensively the current literature describing attempts to develop such inhibitors.

Reviewer #4: Minor proofreading:

1. Introduction, paragraph 3, line 1, correct spelling of ADP-ribosylhydrolase not -layse.

2. In figure 1, a graph from figure 1H is present, but it is not discussed in text and within legend.

3. In figure 4, Figure 4 G and figure 4H, need to be switched from what is actually in the figure.

PLOS authors have the option to publish the peer review history of their article (what does this mean?). If published, this will include your full peer review and any attached files.

Reviewer #1: **Yes: **Jian Zheng

Reviewer #2: No

Reviewer #3: No

Reviewer #4: No
---

## [Editor Report · Decision Letter 1]

16 Aug 2023

Dear Dr. Ott,

We are pleased to inform you that your manuscript 'A single inactivating amino acid change in the SARS-CoV-2 NSP3 Mac1 domain attenuates viral replication in vivo' has been provisionally accepted for publication in PLOS Pathogens.

Best regards,

Jacob S. Yount

Academic Editor

PLOS Pathogens

Sonja Best

Section Editor

PLOS Pathogens

Kasturi Haldar

Editor-in-Chief

PLOS Pathogens

orcid.org/0000-0001-5065-158X

Michael Malim

Editor-in-Chief

PLOS Pathogens

orcid.org/0000-0002-7699-2064
---

## [Editor Report · Acceptance letter]

25 Aug 2023

Dear Dr. Ott,

We are delighted to inform you that your manuscript, "A single inactivating amino acid change in the SARS-CoV-2 NSP3 Mac1 domain attenuates viral replication in vivo," has been formally accepted for publication in PLOS Pathogens.

Best regards,

Kasturi Haldar

Editor-in-Chief

PLOS Pathogens

orcid.org/0000-0001-5065-158X

Michael Malim

Editor-in-Chief

PLOS Pathogens

orcid.org/0000-0002-7699-2064